# Immunosuppressive FK506 treatment leads to more frequent EBV-associated lymphoproliferative disease in humanized mice

Nicole Caduff[1☯], Donal McHugh[1☯], Anita Murer[1], Patrick Rämer[1], Ana Raykova[1], Vanessa Landtwing[1], Lisa Rieble[1], Christian W. Keller[2], Michael Prummer[3], Laurent Hoffmann[4], Janice K. P. Lam[5], Alan K. S. Chiang[5], Friedrich Raulf[4], Tarik Azzi[6], Christoph Berger[6], Tina Rubic-Schneider[4], Elisabetta Traggiai[4], Jan D. Lünemann[2], Michael Kammüller[4], Christian Münz[1]*

1 University of Zurich, Viral Immunobiology, Institute of Experimental Immunology, Zurich, Switzerland, 2 University Hospital of Münster, Department of Neurology with Institute of Translational Neurology, Münster, Germany, 3 Nexus Personalized Health Technologies, ETH Zurich, Zurich Switzerland, and Swiss Institute for Bioinformatics (SIB), Zurich, Switzerland, 4 Novartis Institutes for BioMedical Research, Basel, Switzerland, 5 Department of Paediatrics and Adolescent Medicine, Li Ka Shing Faculty of Medicine, Queen Mary Hospital, The University of Hong Kong, Pokfulam, Hong Kong, 6 Division of Infectious Diseases and Hospital Epidemiology, University Children's Hospital Zurich, Zurich, Switzerland

☯ These authors contributed equally to this work.
* muenzc@immunology.uzh.ch

**Data Availability Statement:** The Ampliseq whole genome profiling raw data and processed datafiles have been deposited in the NCBI Gene Expression

## Abstract

Post-transplant lymphoproliferative disorder (PTLD) is a potentially fatal complication after organ transplantation frequently associated with the Epstein-Barr virus (EBV). Immunosuppressive treatment is thought to allow the expansion of EBV-infected B cells, which often express all eight oncogenic EBV latent proteins. Here, we assessed whether HLA-A2 transgenic humanized NSG mice treated with the immunosuppressant FK506 could be used to model EBV-PTLD. We found that FK506 treatment of EBV-infected mice led to an elevated viral burden, more frequent tumor formation and diminished EBV-induced T cell responses, indicative of reduced EBV-specific immune control. EBV latency III and lymphoproliferation-associated cellular transcripts were up-regulated in B cells from immunosuppressed animals, akin to the viral and host gene expression pattern found in EBV-PTLD. Utilizing an unbiased gene expression profiling approach, we identified genes differentially expressed in B cells of EBV-infected animals with and without FK506 treatment. Upon investigating the most promising candidates, we validated sCD30 as a marker of uncontrolled EBV proliferation in both humanized mice and in pediatric patients with EBV-PTLD. High levels of sCD30 have been previously associated with EBV-PTLD in patients. As such, we believe that humanized mice can indeed model aspects of EBV-PTLD development and may prove useful for the safety assessment of immunomodulatory therapies.

Omnibus (GEO) database under the accession number GSE126515.

**Funding:** This work was supported by grants from the Swiss National Science Foundation (310030_162560 and CRSII3_160708), Cancer Research Switzerland (KFS-4091-02-2017), SPARKS (15UOZ01), the Vontobel Foundation, the Baugarten Foundation, the Sobek Foundation, the Swiss Vaccine Research Institute, the Swiss MS Society, Novartis and the clinical research priority programs on Multiple sclerosis (KFSP MS) and human hemato-lymphatic diseases (KFSP HHLD) of the University of Zurich to C.M. D.M. was supported by an MD-PhD fellowship from the Swiss National Science Foundation (323530_145247). N.C. was supported by a career advancement grant from the University of Zurich (Forschungskredit, FK-18-026). The funders had no role in study design, data collection and analysis, decision to publish, or preparation of the manuscript.

**Competing interests:** The authors have declared that no competing interests exist.

## Author summary

Transplant recipients are medicated with potent immunosuppressive drugs, like FK506, to prevent graft rejection by the host's adaptive immune system. Such treatments can lead to the emergence of post-transplant lymphoproliferative disorders (PTLD) driven by the Epstein-Barr virus (EBV), a ubiquitous human oncovirus that is usually kept under control by T cells. Here, we aimed to establish a model of human EBV-PTLD. To this end, we investigated immunodeficient mice harboring human immune system components reconstituted from human hematopoietic progenitor cells, termed humanized mice. This model enables both the infection of human B cells with EBV and the examination of effects of immunosuppressive compounds on lymphocytes *in vivo*. We found that EBV-associated lymphoproliferations in humanized mice express characteristic viral and human genes observed in EBV-PTLD patients and found similarities in the profiles of serum proteins known for their association with the disease. As such, we believe that EBV-infected humanized mice treated with the immunosuppressive drug FK506 can be used to model specific aspects of EBV-PTLD. Conversely, similar models may prove useful in the preclinical risk-assessment of novel compounds in relation to EBV-associated lymphoproliferation and our study may serve as a template of how one could approach such investigations.

## Introduction

The Epstein-Barr virus (EBV) is one of the most successful pathogens in humans with more than 90% of the adult population persistently infected [1, 2]. At the same time, EBV is a gamma-herpesvirus with potent transforming capacities that can readily immortalize human B cells in culture and drives cancer development in humans. The transformation is achieved by the expression of eight latent viral proteins (six EBV nuclear antigens or EBNAs and two latent membrane proteins or LMPs) and several non-translated RNAs, including two EBV-encoded small RNAs (EBERs) and at least 44 mature microRNAs. This expression pattern is called latency III and is thought to force naïve B cells of healthy virus carriers into differentiation in secondary lymphoid tissues and drive proliferation, thus precipitating post-transplant lymphoproliferative disorder (PTLD) in some individuals after immunosuppressive treatment following transplantation [3–5]. Expression of EBNA1, LMP1 and LMP2 (latency II) rescues differentiating germinal center B cells allowing them to gain access to the memory B cell pool, where EBV persists without the expression of viral proteins (latency 0) or with transient EBNA1 expression (latency I) during homeostatic proliferation [6–8]. Latencies I and II are present in EBV-associated Burkitt's and Hodgkin's lymphoma, respectively. EBV can re-enter the lytic cycle only from latency 0 or I and after methylation of the viral circularized DNA, presumably after antigen-driven plasma cell differentiation [9, 10]. Thus, all EBV expression patterns found in EBV-associated B cell lymphomas are already present in healthy virus carriers and it is thought that comprehensive immune control prevents transition into malignant disease.

Primary immunodeficiencies that predispose for uncontrolled EBV infection point towards cytotoxic lymphocytes as the main components of EBV-specific immune control [2, 11]. Indeed, cytotoxic natural killer (NK) and CD8+ T cells expand during symptomatic primary EBV infection (infectious mononucleosis, IM) [12–14]. Furthermore, NK and T cells contribute to the immune control of EBV infection in mice with reconstituted human immune system components (huNSG mice) as depletion of these cells via monoclonal antibodies leads to

enhanced tumor formation [15–18]. Moreover, the adoptive transfer of EBV-specific T cells transiently controls high viremia in this model [19]. Likewise, adoptively transferred EBV-specific T cells that were expanded *in vitro* can eliminate the pathogenic lymphoproliferation in PTLD patients, emphasizing the importance of T cells in the protection against EBV-transformed B cells [20].

The calcineurin inhibitor FK506 (tacrolimus), which curbs T cell activation, is frequently applied as an immunosuppressant after transplantation [21]. Signaling through the T cell receptor (TCR) after recognition of the cognate antigen presented in the context of MHC molecules is the most important stimulus to activate T cells. TCR signaling leads to the activation of the $Ca^{2+}$-dependent phosphatase calcineurin, which in turn dephosphorylates nuclear factor of activated T cells (NFAT). NFAT may then translocate from the cytoplasm to the nucleus and lead to the transcription of genes such as IL-2 [22]. During FK506 treatment, however, a complex formed by FK506 and the FK506 binding protein interferes with the phosphatase activity of calcineurin and hence TCR-mediated activation of T cells.

Here, we assessed whether humanized mice, which have been used successfully to model facets of human gamma-herpesvirus infection, associated tumor formation and immune control [23], could be used to model aspects of EBV-PTLD. We reasoned that FK506 application would constitute an appropriate component of such a small animal model, since FK506 is commonly administered to prevent allograft-rejection and also carries a known risk for EBV-PTLD development in patients. In our model, treatment with FK506 of HLA-A2 transgenic humanized NSG mice compromised immune control and resulted in increased EBV-associated lymphoproliferations with transcriptional hallmarks of PTLDs. This allowed us to define an EBV-induced transcription pattern in B cells of FK506-treated animals that we exploited to identify biomarkers of uncontrolled EBV infection. Through this process we identified soluble CD30 (sCD30) as a serum marker for the presence of EBV$^+$ tumors in humanized mice. Furthermore, we confirmed high serum sCD30 levels in a cohort of pediatric liver transplant recipients that had developed EBV-PTLD. Importantly, since sCD30 has a known association with EBV-driven PTLD [24], we view the identification of this marker as evidence that aspects of human EBV-PTLD can be recapitulated and that the risk to develop these lymphoproliferations upon immune modulation can be assessed in humanized mice.

## Results

### FK506 treatment of EBV-infected huNSG-A2 mice leads to a higher viral burden and more frequent tumor formation

In order to investigate the effects of FK506 on the course of EBV progression *in vivo*, we reconstituted HLA-A2 transgenic NOD/LtSz-scid IL2Rγ$^{null}$ mice with CD34$^+$ hematopoietic progenitor cells (HPC) from HLA-A2$^+$ human fetal livers (huNSG-A2) (Fig 1A). Upon assessment of human immune cell reconstitution in the peripheral blood at three to four-months after HPC transplantation (S1A and S1B Fig), mice were injected with EBV (strain B95-8) or PBS. Three weeks post-infection (p.i.), animals were either treated with subcutaneous injections of FK506 every second day or left untreated. Apart from intensified grooming at sites of FK506 injection no side effects were observed. Also, FK506 levels were not affected by EBV-infection (Fig 1B). Mice were euthanized five weeks p.i. or earlier if animals displayed symptoms and/or weight loss as specified in the laboratory's animal welfare protocol. Although survival was not significantly decreased during the FK506 administration period, EBV-infected FK506-treated animals that survived until the end of the experiment did experience significantly more weight loss compared to non-treated animals. (S1C and S1D Fig). Accompanying weight loss, EBV-infected FK506-treated huNSG-A2 mice developed significantly

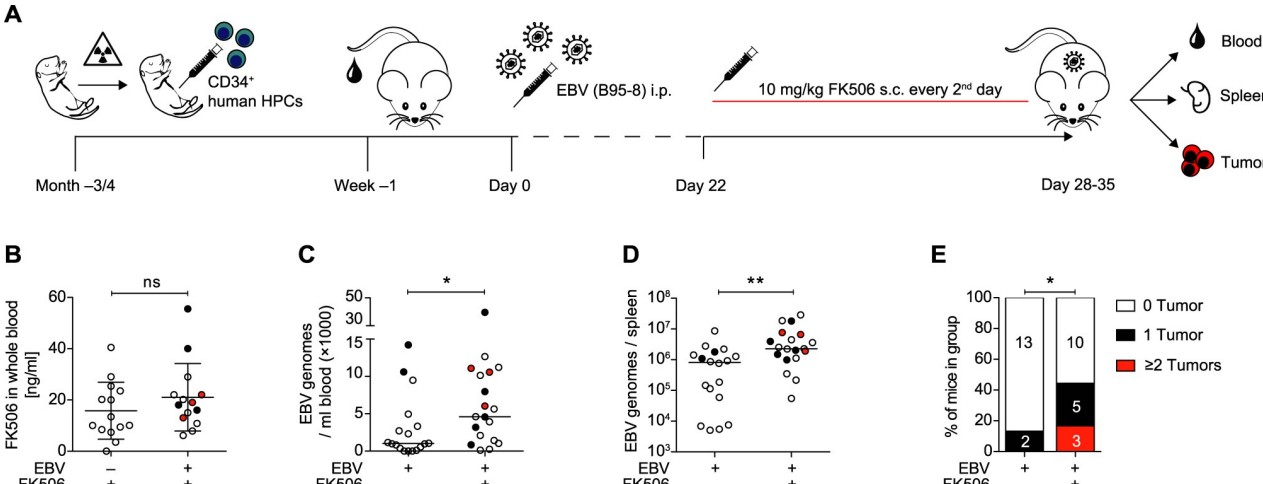

**Fig 1. FK506 treatment of EBV-infected huNSG-A2 mice leads to a higher viral burden and more frequent tumor formation. A)** Experimental set up of huNSG-A2 generation, infection and FK506 treatment. Upon irradiation, NSG-A2 mice received an intrahepatic injection of human CD34+ hematopoietic progenitor cells (HPCs). Reconstitution of human immune system components was assessed in peripheral blood three to four months later. Animals were then injected i.p. with $10^5$ Raji infecting units (RIU) or PBS on day 0. Starting on day 22, animals were treated with FK506 (10mg/kg body weight) or PBS every second day until termination of the experiment. **B)** FK506 concentration was measured 24h after subcutaneous (s.c.) administration of the compound mixture and after a minimum of three applications as described in **A)**. Concentration in whole blood (ng/ml) from individual mice from three independent experiments is depicted. Mean ± SD, p = 0.335 (Mann-Whitney test = MWT). **C)** Quantitative analysis of EBV DNA was performed by qPCR in duplicates for the amplification of BamHI W fragment of EBV in whole blood and **D)** in the spleen of mice that survived at least until day 28. DNA levels were measured at the day of sacrifice and are depicted from five pooled experiments (median, MWT). **E)** Percent of all mice per group with macroscopically visible, EBV-associated tumors, as confirmed via EBNA2 immunohistochemistry (IHC), from four independent experiments. Absolute numbers of mice are indicated per group (tumor score, MWT). **B-D)** Tumor presence in individual mice is indicated by symbol color: clear = no tumors, black = 1 tumor, red = 2 or more tumors. *: p<0.05, **: p<0.01. See also S1 Fig.

higher blood and splenic viral DNA loads and EBV-infected cells were found in the CNS of some animals (Figs 1C, 1D and S1E). The higher viral burden in EBV-infected FK506-treated mice also coincided with a higher frequency of macroscopic tumor formation in the spleen, pancreas, visceral fat, mesentery or liver (Fig 1E). Tumor-bearing and tumor-free mice had similar FK506 levels (S1F Fig). In general, tumor formation was associated with higher blood and splenic viral burden (S1G and S1H Fig), but not with differences in the immune cell reconstitution prior to infection (S1I Fig). Taken together, this indicates that FK506 treatment of EBV-infected huNSG-A2 mice leads to more frequent EBV-induced disease.

## The EBV-induced T cell response is attenuated upon FK506 treatment

Significantly higher numbers of CD8+ T cells at termination in blood and spleen and a significant increase in the blood over time were observed in EBV-infected mice with and without FK506 treatment indicating that the EBV-driven expansion of CD8+ T cells was not affected by the treatment (Figs 2A and S2A–S2C). The concentration of CD4+ T cells was lower in the blood of EBV-infected animals receiving FK506, in line with low total CD4+ T cell counts in PTLD patients [25, 26] (S2I Fig). A similar but non-significant effect on the CD4+ T cell composition was observed in the spleen. To investigate the pattern of expansion in more detail, we characterized the differentiation status of T cells, whereby more central- or effector memory CD8+ T cells were found in EBV-infected animals with and without FK506 treatment (Figs 2B and S2D). The slightly lower level of CD4+ T cells observed in the blood of EBV-infected FK506-treated animals in comparison to their non-treated EBV-infected counterparts was likely due to a lack of expansion of CD4+ T cell memory compartments. FK506 clearly diminished the number of activated HLA-DR+ T cells in non-infected mice (Figs 2C and S2E).

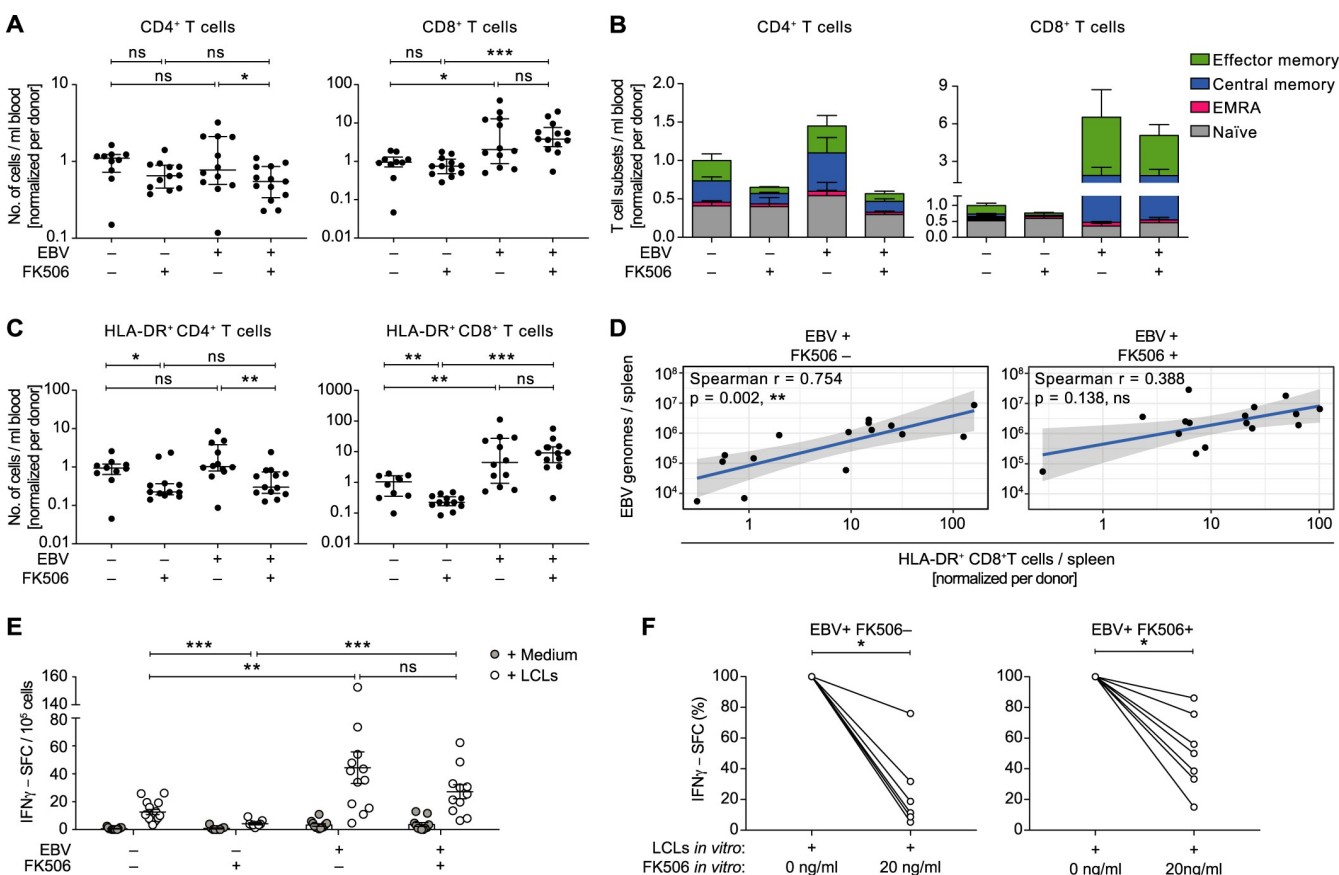

**Fig 2. The EBV-induced T cell response is attenuated upon FK506 treatment. A)** Total peripheral blood CD4+ or CD8+ T cell counts at the end of the experiment from individual mice (n = 10–13 animals per group) are shown relative to the mean of the EBV−FK506− group of each HPC donor. Median (IQR), t test and MWT, respectively. **B)** Differentiation status of T cells in the blood at the day of sacrifice is represented as stacked bar graphs (n = 7–10 animals per group). Naïve: CD62L+ CD45RA+, central memory: CD62L+ CD45RA−, effector memory: CD62L− CD45RA−, effector memory RA+ (EMRA): CD62L− CD45RA+. **C)** Relative numbers of activated CD4+ or CD8+ T cells in peripheral blood were determined by HLA-DR+ surface staining and are depicted for individual mice (n = 10–14 animals per group). Median (IQR), MWT. **D)** Correlations between the splenic EBV DNA load and the relative number of splenic HLA-DR+ CD8+ T cells are depicted for FK506-treated and non-treated EBV-infected mice. r, spearman correlation. Solid lines represent trend lines obtained by linear regression and shaded areas indicate 95% CI of each trend line. **E)** Number of EBV-reactive T cells was assessed for individual mice by Elispot assays whereby IFNγ spot-forming cells (SFC) were counted upon co-culture of CD19 MACS-depleted splenocytes or bone-marrow cells with autologous LCLs or medium. Mean ± SEM, MWT. **F)** Reduction in IFNγ–SFC (%) upon *in vitro* supplementation of 20ng/ml FK506 to CD19 MACS-depleted splenocytes during the co-culture with autologous LCLs. Depicted for EBV-infected mice with (n = 7) and without (n = 6) *in vivo* FK506-treatment. Wilcoxon matched-pairs test on raw IFNγ–SFC counts. Composite data from two (**B**), three (**A, C**) or four (**D, E**) independent experiments. See also S2 Fig. *: p<0.05, **: p<0.01, ***: p<0.001.

However, in EBV-infected mice, FK506 treatment only restrained the expansion of activated CD4+ T cells but not activated CD8+ T cells in the blood. Consistent with an interrupted TCR signaling pathway, the serum levels of IL-2 were decreased in FK506-treated compared to non-treated animals with or without EBV infection (S2H Fig). This mirrors the lower activated CD4+ T cell count in treated mice, since CD4+ T cells represent the main producers of IL-2 and in turn also depend in part on IL-2 signaling for survival [27].

Upon sole EBV infection, activated and total CD8+ T cell numbers correlated positively with the EBV titers in the spleen, whereas in EBV+ FK506+ mice no such correlations could be observed (Figs 2D and S2J). However, within this group, tumor-bearing mice, which had similar levels of CD4+ T cells, had higher numbers of (activated) CD8+ T cells in the spleen compared to tumor-free mice (S2F and S2G Fig). In order to estimate the frequency of EBV-

specific T cells, we assessed the number of IFNγ-secreting cells upon contact with autologous EBV-transformed B cells (LCLs) in individual mice from the different groups. We found similar rates of LCL-reactive cells in EBV-infected mice with and without FK506 treatment upon *ex vivo* co-culture (Fig 2E). We postulated that the *in vivo* suppressive effect of FK506 on EBV-reactive T cell function was not retained *ex vivo* due to washing out of the compound during cell isolation, as previously reported for human samples [28]. Indeed, upon supplementation of FK506 at a concentration in line with the compound levels measured *in vivo* (20ng/ml), the EBV-specific response was significantly lower compared to culture conditions with carrier solution supplementation (Figs 2F and S2K). As such, despite similar rates of LCL-reactive cells in FK506-treated and non-treated EBV-infected animals measured *ex vivo*, these cells were likely attenuated *in vivo*.

Taken together, FK506 treatment seems to inhibit the activation, differentiation and proliferation primarily of CD4+ T cells in this model, whereas EBV promotes CD8+ T cell expansion even in the presence of the immunosuppressive compound. However, virus-dependent CD8+ T cell activation and expansion was likely curbed by FK506 treatment *in vivo* and/or the associated loss of CD4+ T cell help, even though high overall rates of activation and expansion were achieved during EBV infection.

### FK506 treatment enhances the expression of EBV latency II/III transcripts in B cells in vivo

EBV-driven PTLD in patients displays latency II or III associated gene expression [1]. In order to characterize the viral latency pattern upon FK506 treatment *in vivo*, we analyzed the EBV-specific gene expression in CD19+ splenocytes. Firstly, we found that FK506 treatment does not influence the abundance of B cells in the spleen with or without EBV infection (Fig 3A) and B cell counts were similar between tumor-bearing and tumor-free mice of the EBV+ FK506+ group (S3A and S3B Fig). Furthermore, FK506 did not affect the frequency of detection or the relative expression of the latency I/II-associated *Qp-EBNA1* or the immediate early lytic transcript *BZLF1*, respectively (Fig 3B and 3C). When we analyzed the expression of EBV latent transcripts associated with the transforming latency III (*Cp/Wp-EBNA1*, *EBNA2*) and II/III (*LMP1*, *LMP2A*), we observed significantly increased expression levels of all of these viral oncogenes in B cells of FK506-treated compared to untreated huNSG-A2 mice whereby the highest levels were noted among tumor-bearing mice (Fig 3D). Elevated transcript levels could be due to a higher frequency of infected cells or increased levels of transcription per infected cell. To investigate this, we normalized the transcript levels to *EBER1*, a small non-coding EBV RNA present in all latency patterns. No general difference in the expression of latency III transcripts was detected between the two groups upon *EBER1* normalization. Furthermore, no differences were apparent between tumor-bearing and tumor-free mice (S3C and S3D Fig). Hence, the increased presence of latency II/III transcripts is likely a result of enhanced accumulation of EBV-transformed B cells, rather than a shift in latency patterns. In line with elevated *EBNA2* transcript levels, we found more EBNA2+ cells in the spleen of FK506-treated mice (Fig 3E). We view the enhanced expression of latency III transcripts in the B cell compartment as further evidence of a loss of EBV-specific immune control during FK506 treatment, allowing EBV-infected B cells to expand and continue to express viral antigens, which promote B cell transformation and tumor formation.

### Host transcriptome profiling reveals differential gene expression in B cells of EBV-infected huNSG-A2 mice treated with FK506

In order to investigate the effect of the treatment with FK506 and the associated higher EBV latent gene expression on the B cell host transcriptome, we performed whole transcriptome

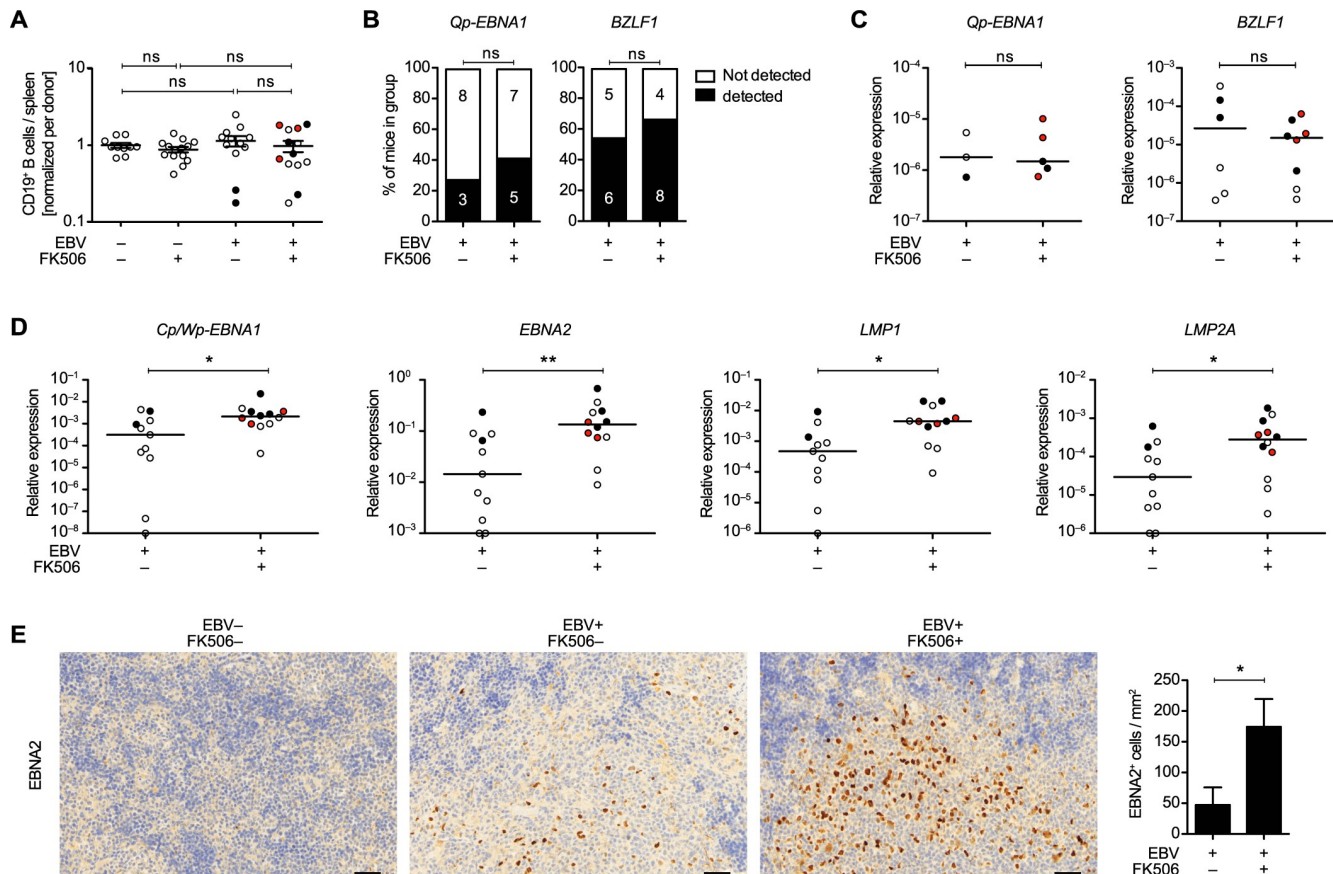

**Fig 3. FK506 treatment enhances the expression of EBV latency II/III transcripts in B cells *in vivo*. A)** Total splenic CD19$^+$ cell counts are depicted for individual mice from three independent experiments relative to the mean of the EBV$^-$FK506$^-$ group. Mean ± SEM, unpaired t test with Welch's correction. **B-D)** Composite data from three independent experiments with EBV$^+$ FK506$^-$ (n = 11) and EBV$^+$ FK506$^+$ (n = 12). **B)** EBV mRNA in CD19$^+$ MAC-sorted splenocytes was quantified in triplicates for each mouse by RT-qPCR. Data is depicted as percent of mice per group with transcript levels above the detection threshold. P-values were calculated with Fisher's exact test. **C-D)** Transcript expression is displayed relative to the two reference genes (geometric mean of *GAPDH* and *SDHA*). EBNA, EBV nuclear antigen; LMP, latent membrane protein; Cp, Wp and Qp are latency-specific EBV promoters. Transcript values below the reliable quantification level are plotted on the X-axis. **E)** Representative IHC stainings and quantification of EBV nuclear antigen 2 (EBNA2) in splenic sections from EBV$^-$FK506$^-$, EBV-infected (n = 7) and EBV-infected, FK506-treated (n = 10) huNSG-A2 mice from two independent experiments. Scale bar: 50µm. P-values were calculated with MWT, *: p<0.05, **: p<0.01. See also S3 Fig.

AmpliSeq RNA profiling of CD19-purified splenocytes (S4 Fig). Principal Component Analysis (PCA) of the gene transcription profiles revealed clustering of the sampled mice based on the experimental groups (Fig 4A). The largest percentage of the variance in the data was due to the presence of EBV (Fig 4A, PC1), as such the comparisons of EBV-infected with the non-infected control groups gave rise to the majority of differentially expressed genes (DEGs) (Fig 4B). Furthermore, hierarchical clustering of the 100 most strongly regulated DEGs grouped the mice according to their experimental conditions, indicating distinct gene transcription patterns in the B cells from the four different groups (S5 Fig). Genes previously associated with EBV infection of B cells *CD28* [29] and *CXCL10* [30, 31] were up-regulated in EBV-infected groups, whereas the PTLD-associated transcripts *IL6* [25] and *TNFSFR8* (CD30) [32] were up-regulated specifically in the B cells from EBV-infected FK506-treated animals (Fig 4C).

Based on the DEGs in the experimental groups we performed competitive gene set enrichment analysis (GSEA) with a curated selection of 638 gene sets from the MSigDB (S1 Table). Despite the high number of individual DEGs in the EBV$^+$ FK506$^-$ vs. EBV$^-$FK506$^-$

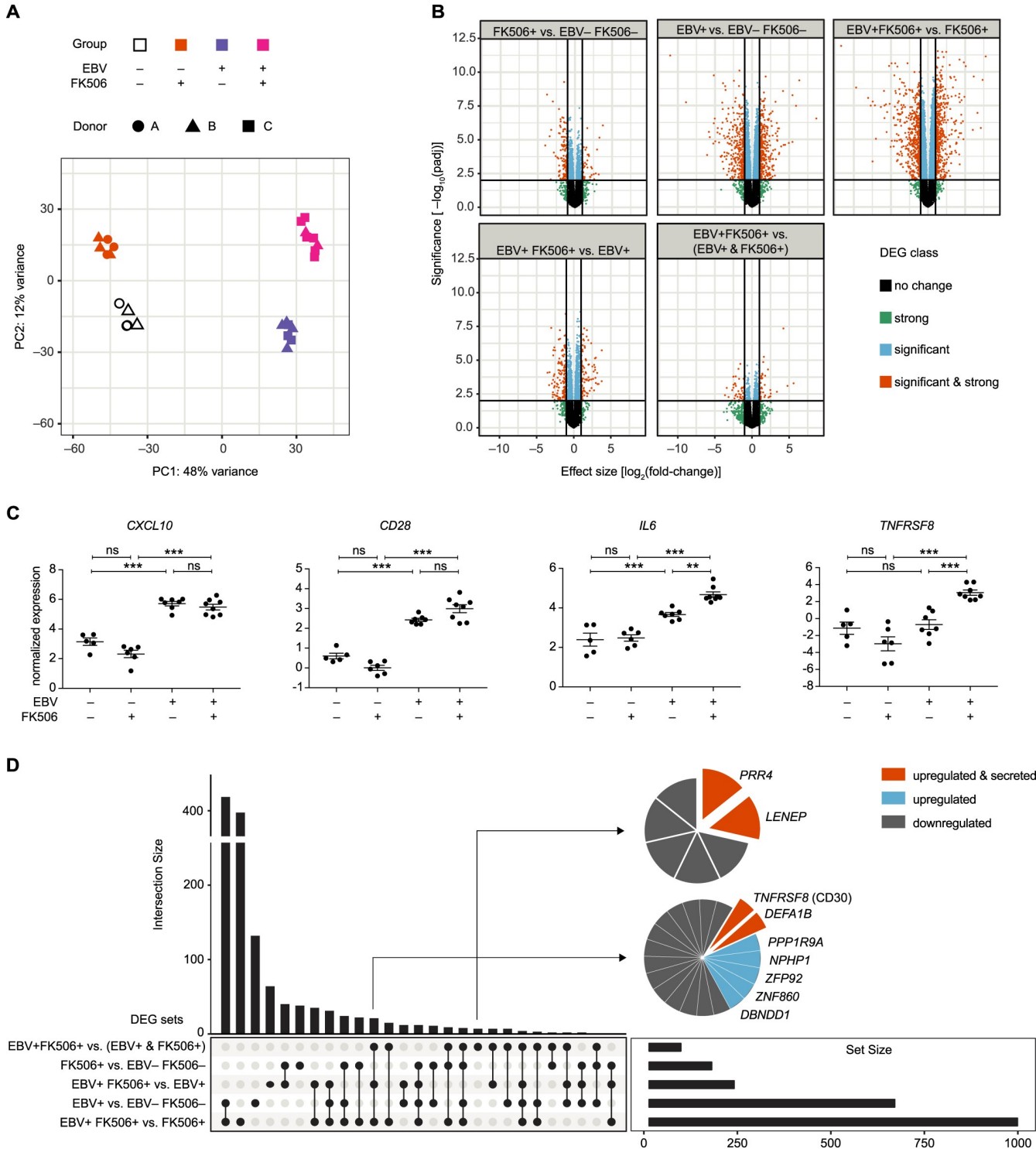

**Fig 4. Host transcriptome profiling reveals differential gene expression in B cells of EBV-infected huNSG-A2 mice treated with FK506. A)** RNA expression profiling of the human B cell transcriptome of mice from three independent experiments with EBV⁻FK506⁻ (n = 5, clear), EBV⁻FK506⁺ (n = 6, orange), EBV⁺ FK506⁻ (n = 7, purple) and EBV⁺ FK506⁺ (n = 8, magenta). Principal component plot: Sample similarity in a 2D projection by multi-dimensional scaling. Spurious batch effects of unknown origin and HFL donor-specific effects were corrected by surrogate variable analysis. The two principal components represent 60% of the total variance in the data set. **B)** Volcano plots: The horizontal line corresponds to FDR = 0.01 and genes below this line are labeled 'not significant'. The two vertical lines correspond to a 2-fold change in expression and genes outside this range are labelled 'strong'. The legend refers to 'not'—not significant and weak effect; 'signif & strong'—significant and strong effect; 'signif'—significant but weak effect; 'strong'—not

significant but strong effect. **C)** Normalized AmpliSeq expression levels of *CXCL10*, *CD28*, *IL6* and *TNFRSF8* (CD30) are depicted for individual mice with mean ± SEM and FDR-corrected p-value summaries (*: FDR-p<0.05, **: FDR-p<0.01, ***: FDR-p<0.001.) **D)** Upset plot identifies private, shared and promiscuous DEGs among groups. The vertical bar chart indicates the intersection size of transcripts shared between certain groups defined by solid points below the chart. The total number of DEGs for each condition is depicted in the horizontal bar chart. See also S4–S6 Figs and S1 and S2 Tables.

comparison, we did not identify many significantly enriched gene sets (S6A Fig, S2 Table). We assume that B cell gene sets were not uniformly up- or down-regulated, since the B cell population from EBV-infected mice contained not only EBV-infected but also many non-infected cells nevertheless responding to the ongoing infection of the host with altered transcription. Hallmark IFNγ target genes were down-regulated in EBV⁻FK506⁺ vs. EBV⁻FK506⁻ mice, in line with significantly lower IFNγ levels in the serum of FK506-treated mice (S6B Fig). Conversely, IFNγ targets were up-regulated in EBV⁺ FK506⁺ vs. EBV⁻FK506⁺ mice, in line with significantly higher levels of IFNγ in EBV⁺ FK506⁺ mice thus validating the GSEA results. Although certain gene sets indicating enhanced c-myc activity were up-regulated under the influence of FK506 alone, more myc-specific signatures were identified upon comparison of EBV⁺ FK506⁺ and EBV⁺ FK506⁻ mice. This could be a consequence of more frequent EBNA2⁺ cells (Fig 3D and 3E) since EBNA2 drives c-myc expression during the early phase of EBV infection [33]. Moreover, within the EBV⁺ FK506⁺ vs. EBV⁺ FK506⁻ comparison, we could document an enrichment of genes transcribed as a result of LMP1 expression, coinciding with the enhanced *LMP1* expression found in the B cell fraction (Fig 3D). Higher LMP1 expression may also explain enhanced transcription of cell cycle related hallmark E2F targets since LMP1 induces mitogenic B-cell activation through c-myc and Jun/AP1 family members [33]. Furthermore, gene sets indicating plasma cell differentiation were down-regulated in the EBV⁺ FK506⁺ vs. EBV⁺ FK506⁻ comparison as a possible consequence of the EBV latency III genes EBNA3A and 3C mediated repression of plasma cell differentiation upon infection and latency establishment [34]. Therefore, EBV induced a distinct transcriptional profile in B cells of FK506-treated hosts. The differential gene expression dataset allowed us to discriminate sets of DEGs that are significantly affected by FK506 treatment, EBV infection and DEGs that are only differentially regulated within the EBV⁺ FK506⁺ group compared to sole EBV infection or FK506 treatment (Fig 4D). We selected DEGs contained in these sets for their potential as biomarkers for uncontrolled EBV infection upon FK506 treatment, based on the probability to detect them in the plasma upon peripheral blood sampling. We identified PRR4, soluble (s) CD30 (gene TNFRSF8), LENEP and DEFA1B as candidate biomarkers, of which the former three were selected for further investigation as they were up-regulated in the EBV⁺ FK506⁺ group and described to be secreted. DEFA1B was not further evaluated, as RNA transcripts were not detectable in over 45% of the mice tested.

## High serum levels of sCD30 are associated with EBV-driven tumor presence in huNSG-A2 mice and in human PTLD patients

In order to investigate the potential of the identified candidate biomarkers *in vivo*, serum levels of PRR4, sCD30 and LENEP were assessed via ELISAs. Both PRR4 and sCD30 were detectable at higher levels in EBV⁺ FK506⁺ mice compared to non-infected FK506-treated animals (Figs 5A and S7A). The concentration of LENEP was low in general and below the quantification threshold in 50% of animals across all groups (S1 Data). Interestingly, mice with tumors had higher levels of sCD30 and PRR4 compared to (EBV⁺ or−) FK506-treated animals without tumors (Figs 5B and S7B). However, investigation of the serum of pediatric liver transplant recipients indicated that PRR4 levels would likely not serve as a useful biomarker for detecting PTLD (S7C Fig). In contrast, levels of sCD30 were significantly higher in PTLD patients at or

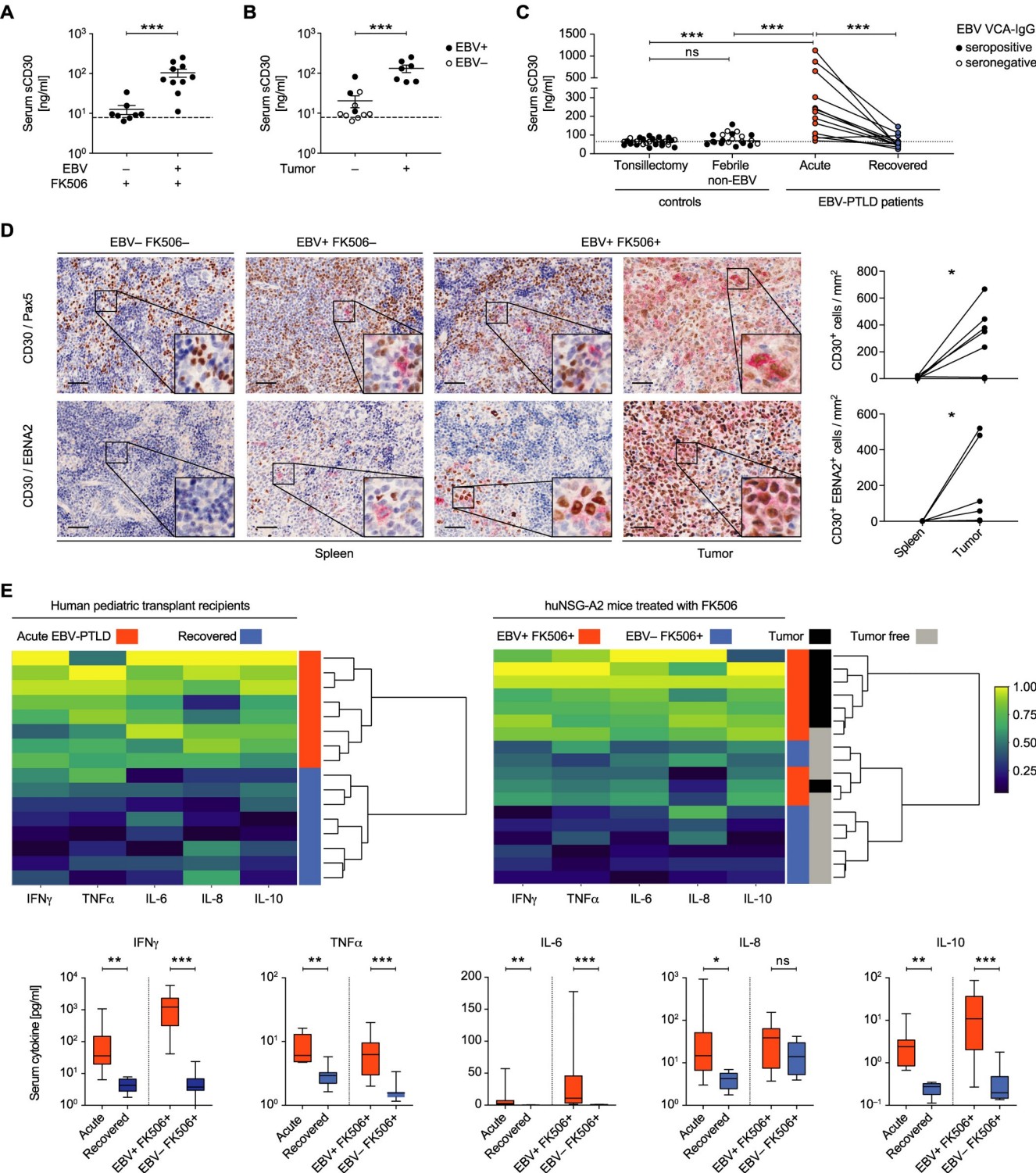

**Fig 5. High serum levels of sCD30 are associated with EBV-driven tumor presence in huNSG-A2 mice and in human PTLD patients. A-C)** Protein concentration of sCD30 was measured in serum samples with ELISA. **A)** Serum was obtained at the day of sacrifice from animals from two independent experiments. **B)** *Post hoc* stratification of FK506-treated mice with or without macroscopically visible tumors with EBV-infection status indicated by black (EBV⁺) or clear (EBV⁻) symbols. **A-B)** Mean ± SEM, MWT. Dashed lines indicate quantification thresholds. **C)** sCD30 was measured in the sera of 31 healthy children undergoing elective tonsillectomy, 21 children that presented with non-EBV associated fever (Febrile non-EBV, EBV VCA-IgM negative) and 13 pediatric PTLD patients around diagnosis (PTLD Acute) and again at least 36 months later (PTLD Recovered) (See S3 and S4 Tables). Dashed line represents the median of the healthy controls. MWT and Wilcoxon matched-pairs test. **D)** Representative immunohistochemistry staining of CD30 (red) and

PAX5 (brown), or CD30 (red) and EBNA2 (brown), in splenic sections or tumor tissue of huNSG-A2 mice. Scale bars: 50µm. Quantification of CD30$^+$ (n = 7) and CD30$^+$ EBNA2$^+$ cells (n = 6) per mm$^2$ in tumor sections and non-tumorous spleen tissue of EBV-infected mice is depicted for two independent experiments. **E)** Cytokine concentration was measured in the serum of EBV-PTLD patients (n = 8) at diagnosis (acute) and again at recovery time point at least 36m post-diagnosis and in FK506-treated EBV$^-$ (n = 8) and EBV$^+$ (n = 10) experimental mice, composite data from two independent experiments. Upper panel: Heatmaps and hierarchical clustering of pediatric patients or huNSG-A2 mice based on serum cytokine levels (IFNγ, TNFα, IL-6, IL-8 and IL-10) at diagnosis (red) and recovery (blue) for PTLD (left) and on the day of sacrifice for huNSG-A2 mice (right) treated with FK506 (blue) and additionally EBV-infected (red). Tumor presence versus absence in huNSG-A2 mice is indicated in black and grey respectively. Lower panel: Cytokine concentration (median (IQR) with min and max range as whiskers) in the serum of EBV-PTLD patients and in FK506-treated EBV$^-$ and EBV$^+$ experimental mice. Wilcoxon matched-pairs test and MWT respectively. $^*$P < 0.05, $^{**}$P < 0.01, $^{***}$P < 0.001. See also S7 Fig and S3 and S4 Tables.

shortly after diagnosis compared to the same individuals upon recovery (Fig 5C and S3 Table). In control cohorts of healthy children undergoing elective tonsillectomy and children that presented with acute non-EBV-associated febrile illnesses the serum concentration of sCD30 was relatively low. Furthermore, sCD30 levels were similar between the EBV VCA-IgG seropositive and seronegative controls (S7D Fig and S4 Table). In line with higher sCD30 serum concentration, co-stainings of PAX5 and EBNA2 with CD30 revealed more CD30-expressing cells and CD30/EBNA2 co-expressing cells, respectively, within tumors compared to non-tumorous spleen tissue of huNSG-A2 mice (Fig 5D). Furthermore, these EBV-transformed cells retain the ability to produce high levels of sCD30 when explanted from EBV-infected humanized mice and expanded as immortalized LCLs *in vitro* (S7E Fig).

The discrepancy regarding PRR4 as a marker in humanized mice versus human patients may result from the fact that the hematopoietic system is the only source of human PRR4 in humanized mice, while in human patients PRR4 from other sources, such as the salivary and lacrimal glands, may obscure PTLD-specific patterns [35]. This emphasizes an expected limitation of the huNSG-A2 system, as human disease-associated gene expression of non-hematopoietic origin cannot be modelled. In order to further characterize PTLD-associated markers of hematopoietic origin, we investigated the inflammatory cytokine profiles in the sera of PTLD patients at diagnosis and after recovery and compared these with FK506-treated humanized mice with and without EBV infection. Unsupervised hierarchical clustering grouped patients at diagnosis and after recovery into two main clusters based on five cytokines (Fig 5E). A similar clustering was achieved when analyzing the humanized mice, especially when highlighting those with macroscopic tumor formation, which overall had higher concentrations of these serum cytokines. Elevated levels of IL-6 and -10 *in vivo* may be partially explained by the secretion from EBV$^+$ B cells themselves, as LCLs derived from EBV-infected humanized mice readily produce IL-6 and especially IL-10 in culture (S7F Fig).

In summary, we identified (s)CD30 as a biomarker for tumorigenesis during FK506-mediated immunosuppression via the investigation of EBV-infected humanized mice. (S)CD30 has a known association with EBV-driven PTLD [24, 32] and has also been successfully employed as a therapeutic target in relapsed Hodgkin's lymphoma and systemic anaplastic large cell lymphoma [36]. We view sCD30 accumulation in FK506-treated, tumor-bearing mice along with the additional similarities in the inflammatory cytokine profile between humanized mice and PTLD patients as confirmation of the suitability of humanized mice to model immune system-specific aspects of PTLD. Furthermore, this combined soluble biomarker profile could further be investigated in clinical samples as a predictive signature for PTLD development in solid organ transplant recipients under FK506-based maintenance therapy. As such, this mouse model could be used to recapitulate aspects of EBV-PTLD biology or to assess risk of uncontrolled EBV infection with novel immunosuppressive compounds in comparison to FK506.

## Discussion

The use of potent immunosuppressive drugs in transplant recipients markedly increases the risk of EBV-associated PTLD, which is often fatal if not recognized and treated. Our study demonstrates that mice with human immune system components can be used to model EBV-driven tumor formation upon the administration of the immunosuppressant FK506. We find that FK506 primarily affects CD4[+] T cell activation in the presence of EBV infection. This led to markedly reduced IL-2 production and compromised immune control of the B cell transforming EBV latency III infection, resulting in elevated viral loads and tumor formation akin to PTLD.

The emergence of HLA class I-presented EBV latent antigen-specific CD8[+] T cells in patients is associated with long-term control of the virus during IM. IFNγ-producing CD8[+] T cells responding particularly to the antigens expressed during latency III, such as EBNA3A-C, tended to expand as the cellular and plasma viral load dropped over time after primary EBV infection [37]. However, EBV-specific immune control by T cells is exquisitely sensitive to NFAT-mediated regulation of IL-2 production, which is inhibited by FK506. IL-2 producing CD4[+] T cells are required to maintain functional CD8[+] T cell-mediated immune control of persistent infection [38]. Indeed, individuals with EBV-associated autosomal lymphoproliferative syndrome, a primary immunodeficiency characterized by uncontrolled EBV infection, have mutations in the IL-2 inducible T cell kinase (ITK) [39–46], which is required during T cell receptor signaling as it elicits the phospholipase Cγ1 (PLCγ1)-mediated $Ca^{2+}$ release upstream of the calcineurin-NFAT pathway [22]. To date, a total of thirteen patients with loss-of-function mutations in ITK have been reported, twelve of which presented with EBV-positive lymphoproliferations (as reviewed by Tangye et al. [11]). Interestingly, these patients also exhibited CD4[+] T cell lymphopenias, which in one case was already present prior to EBV infection [46]. Furthermore, two additional primary immunodeficiencies that compromise T cell receptor signaling present with CD4[+] T cell lymphopenia and uncontrolled EBV infection, as observed under FK506 treatment of EBV-infected humanized mice. One immunodeficiency affects cytosolic $Mg^{2+}$ levels, required for PLCγ1 activation, via mutations in the magnesium transporter 1 (MAGT1) in patients with X-linked immunodeficiency with magnesium defect, EBV infection, and neoplasia (XMEN) [47–51]. The other immunodeficiency is caused by mutations in the guanine nucleotide exchange factor RasGRP1, downstream of PLCγ1 signaling [52, 53]. Similarly, our pharmacological interference with T cell receptor signaling via the calcineurin inhibitor FK506 led to CD4[+] T cell lymphopenia with and without EBV infection and compromised immune control of EBV titers as well as associated lymphoproliferative diseases. Thus, mice with reconstituted human immune system components recapitulate the sensitivity of EBV-specific cell-mediated immune control to immune-modulatory therapies that affect T cell receptor signaling.

The resulting EBV-associated lymphoproliferations presented with certain transcriptional hallmarks associated with PTLD such as high expression levels of IL-6 and CD30 [25, 32]. Both have been implicated in lymphoproliferations by mediating B cell activation through STAT3 and NFκB, respectively [54, 55]. Accordingly, they are therapeutically targeted in some B cell malignancies, like Castleman disease and Hodgkin's lymphoma [54, 55]. Similarly, previous studies have shown that EBV infection of humanized mice can generate tumors that, although not identical to the human malignancies, indeed recapitulate certain transcriptional hallmarks of EBV-associated diffuse large B cell lymphomas (DLBCLs) and primary effusion lymphomas (PELs) [30, 56]. EBV deficient in its nuclear antigen 3B (EBNA3B) was detected in a subset of patients with DLBCL [30, 57]. Infection of huNSG mice with EBNA3B-deficient EBV (ΔEBNA3B EBV) recapitulated increased lymphomagenesis with low lymphocyte

infiltrates into the tumor microenvironment, reminiscent of DLBCLs. Furthermore, transcriptional similarities of *in vivo* ΔEBNA3B EBV-transformed B cells to patient-derived tumor cell lines were detected, including an inability to produce pro-inflammatory chemokines that when reintroduced into these lymphoma cell lines restored their T cell-mediated immune control [30]. Furthermore, co-infection of EBV with its closely related Kaposi sarcoma-associated herpesvirus (KSHV) led to increased tumorigenesis in huNSG mice with transcriptional hallmarks of PELs, a rare human tumor frequently dual-infected with EBV and KSHV [1, 56]. Specifically, EBV and KSHV co-infection of huNSG mice recapitulated PEL-associated plasma cell differentiation and increased lytic EBV reactivation in human tumors [56, 58]. Due to these similarities of EBV-associated lymphomas in huNSG mice and patients, the transcriptional characteristics of PTLD-like tumors of FK506-treated EBV-infected mice could be further explored for biomarkers of PTLD development in transplant patients, as shown for sCD30 and IL-6 in the current study. This comparison may allow the discrimination between transcriptional changes in B cells upon EBV-associated lymphoproliferation and those due to the transplantation requiring morbidity in patients. We found striking similarities in EBV-associated disease formation between mice with reconstituted human immune system components and patients upon modification of T cell receptor signaling by either pharmacological modulation or mutation, respectively. This further suggests that novel immune-modulatory treatments can be explored for their risk to predispose for PTLD development during EBV infection of mice with reconstituted human immune system components.

## Materials and methods

### Ethics statement

Animal protocols were reviewed and approved by the veterinary office of the canton of Zurich, Switzerland (protocol 209/2014). The use of human fetal liver-derived HPCs was approved by the cantonal ethics committee of Zurich (KEK-ZH-Nr. 2010–0057). Studies using serum samples of EBV-PTLD donors or from febrile (non-EBV) patients were reviewed and approved by the Institutional Review Board of the University of Hong Kong / Hospital Authority Hong Kong West Cluster (IRB reference numbers UW16-508 and EC1940-02, respectively). Studies using serum samples of healthy children undergoing elective tonsillectomy at the University Children's Hospital of Zurich were reviewed and approved by the cantonal ethics committee of Zurich (KEK-ZH Nr. St 40/05). Informed consent was obtained from the pediatric patients' parents or legal guardians prior to study inclusion.

### HuNSG-A2 generation, infection and treatment

HLA-A2 transgenic NOD/LtSz-scid IL2Rγ$^{null}$ (NSG-A2) mice were obtained from the Jackson Laboratories, bred and maintained at the Institute of Experimental Immunology, University of Zurich. Newborn NSG-A2 mice (up to five days old) were irradiated with 1 Gy. Five to seven hours after irradiation, mice were injected intrahepatically with $1–4×10^5$ CD34$^+$ human hematopoietic progenitor cells (HPCs) derived from human fetal liver (HFL) tissue obtained from Advanced Bioscience Resources. Isolation of human CD34$^+$ cells from HFL tissue was performed as previously described by positive selection for CD34 with magnetic cell separation according to the manufacturer's recommendations (Miltenyi Biotec) [15, 30]. HLA typing of HFL-derived HPCs was performed by a PCR sequence-specific oligonucleotide reverse assay using commercial HLA kits (Fujirebio Diagnostics Inc.). Individual mouse cohorts were reconstituted with cells derived from five different HLA-A2$^+$ HFL donors. Reconstitution of human immune system components in mice was analyzed three months after human CD34$^+$ cell injection and again if necessary in the week before the start of the experiments by flow

cytometric immune phenotyping of PBMCs for huCD45, huCD3, huCD19, huCD4, huCD8, huNKp46 as previously described by McHugh et al. [56]. Apart from bleeding for analysis of human immune cell reconstitution in peripheral blood, mice were not involved in any procedures prior to viral infection. Average huCD45$^+$ cells of peripheral blood lymphocytes before experiment was 82.7% ± 11.4%, huCD3$^+$ T cells of huCD45$^+$ 27.5% ± 13.6%, huCD19$^+$ B cells of huCD45$^+$ 61.2% ± 14.6%, huCD4$^+$ T cells of human T cells 70.1% ± 10.6%, huCD8$^+$ T cells of human T cells 26.8% ± 9.8%, CD3$^-$ NKp46$^+$ NK cells of human lymphocytes 2.3% ± 1.2%, (Mean ± SD, n = 80, 42 female and 38 male mice). Recombinant EBV B95-8 (2089) was produced in HEK 293 cells as described previously [59] and virus titers were determined by flow cytometric analysis of GFP-positive Raji cells 48 hours post infection. 13 to 19 weeks after engraftment, huNSG-A2 mice were injected with 10$^5$ RIU of EBV or 100μl PBS intraperitoneally. Animals were treated with 10mg/kg body weight of FK506 (Prograph injectable solution, Astellas) diluted 1:3 in PBS subcutaneously every second day starting at day 22 p.i. or left untreated. Side effects of FK506 treatment: intensified grooming or scratching at sites of injection was observed. Each experiment was performed with a cohort of mice reconstituted with CD34$^+$ cells from a single donor and animals were distributed to the experimental groups to ensure a similar ratio of males to females and overall similar human immune reconstitution in the peripheral blood. Animals were sacrificed five weeks post infection or euthanized earlier if necessitated by the laboratory's animal welfare protocol due to general health symptoms and weight loss over 15%. Investigators were not blinded regarding the virus vs. Mock-infection of the animals. EBV-inoculated mice with no EBV DNA load in blood and spleen and no EBV nuclear antigen 2 staining in FFPE splenic sections were considered non-infected and analyzed together with PBS-injected animals in the respective non-infected group designated as "EBV$^-$FK506$^-$"and "EBV$^-$FK506$^+$". This was the case in 5 out of 25 mice injected with EBV and 4 out of 28 injected with EBV and treated with FK506 (two-sided Fisher's exact test, p = 0.72). The tumor frequency in infected huNSG-A2 mice was assessed by a tumor score: no tumors observed = 0; tumor observed = 1; multiple tumors observed = 2.

## AmpliSeq transcriptome profiling

Single cell suspensions of huNSG-A2-derived splenocytes were subjected to lysis of erythrocytes by ACK lysing buffer (Life Technologies). B cells were purified by magnetic cell sorting using CD19 microbeads (Miltenyi Biotech) according to the manufacturer's recommendations. Total RNA of CD19$^+$ cells (10$^6$–10$^7$ cells) was isolated using the RNeasy Mini Kit (QIAGEN) and traces of genomic DNA were removed by on-column DNase digestion with the RNase-free DNase set (Qiagen). Purified RNA was stored at −20°C. All mice for which B cell RNA was isolated and RNA of sufficient amount was available after EBV transcript RT-qPCR were eligible for Ampliseq profiling (n = 31). The five mice with the lowest levels of *EBER1* were not included in the analysis. This resulted in 5, 6, 7 and 8 samples for the four respective conditions (total = 26). RNA quality was assessed further by an Agilent 2100 Bioanalyzer using the RNA 6000 Nano Kit. All RIN numbers were ≥ 9.5. RNA was quantified by NanoDrop 1000 spectrophotometry. Reverse transcription of 10 ng total RNA, amplification of targets by ultra-high multiplex PCR, and sequencing library construction was performed using the Ion AmpliSeq Transcriptome Human Gene Expression Kit (Thermo Fisher Scientific; #A26325) and Ion Xpress barcodes. Unamplified libraries were quantified by qPCR and equal amounts were combined for sequencing (6 μl of 100 pM total libraries). Templating of Ion Sphere Particles by emulsion PCR was conducted using the Ion 540 Kit (Thermo Fisher Scientific #A27753) and an OT2 instrument. Sequencing using an Ion 540 semiconductor chip was carried out on the Ion S5 System (Thermo Fisher Scientific). Initial analysis, quality control, and

normalization of mapped reads as RPM (reads per million mapped reads) was performed by the ampliSeqRNA plug-in on the Ion Torrent server. Average number of mapped reads for all samples was $10.307 \times 10^6$, minimum and maximum were $3.746 \times 10^6$ and $29.005 \times 10^6$ mapped reads, respectively. The library size of the 26 samples was normalized during preprocessing to 1 million. All subsequent statistical analysis of the transcriptome data was performed using the R environment for statistical computing (R version 3.4.1 (2017-06-30)) with Bioconductor [60] and dedicated packages. To avoid technical artifacts the data were filtered such that genes for which the average over all samples was less than 1 were removed. Identification and correction for spurious batch effects of unknown origin was performed via application of surrogate variable analysis [61]. Differential expression was modeled using mixed-effect linear regression with empirical Bayes variance estimation, as implemented in the limma package (3.34.3) [62]. To account for the repeated measurements of material from the same donor, a subject-specific additive offset was included in the model equation. Mapping of the transcripts to HGNC symbols and ENTREZID's was performed using the org.Hs.eg.db package version 3.5.0. Competitive enrichment analysis of 638 curated gene sets (S1 Table) was performed, where p-values are estimated via random rotations as implemented in the camera function of the limma package [62]. Gene sets were selected from MSigDB with the help of the Walter + Eliza Hall Institute of Medical Research's mouse and human orthologs of the MSigDB in R format and included human gene sets related to B cell differentiation, proliferation and apoptosis. The transcriptome profiling data has been deposited in the Gene Expression Omnibus (GEO) database (https://www.ncbi.nlm.nih.gov/geo/) with the accession number GSE126515. Summary of GSEA results may be found in the Supporting information (S2 Table).

## Statistics

Data with a Gaussian distribution, as determined by the D'Agostino & Pearson omnibus test, were compared with a two-tailed unpaired t test or a two-tailed unpaired t test with Welch's correction if variances were significantly different as determined by the F test unless otherwise stated. Matched data with a Gaussian distribution were compared with the two-tailed paired t test, matched data with a non-Gaussian distribution were compared with the Wilcoxon matched-pairs test. Data with a non-Gaussian distribution were compared with the two-tailed Mann-Whitney test (MWT). Survival curves were compared with the log-rank test. Categorical data were assessed with the two-tailed Fischer's exact test. Correlation was assessed by Spearman's rank test. A p value of $<0.05$ was considered statistically significant. Statistical analysis was performed with Prism (GraphPad) software or R (see AmpliSeq transcriptome profiling).

## Immunohistochemical staining and analysis

Tissue was fixed in 4% buffered formalin, paraffin-embedded (FFPE), followed by deparaffinization and processing on a BOND-MAX automated IHC system (Leica Microsystems). Antigen retrieval for EBNA2 single-stainings was performed in Bond Epitope Retrieval Solution 2 (ER2, Leica) for 30 min at 100°C followed by 30 min incubation with monoclonal mouse anti-EBNA2 (clone PE2, Abcam) and detection using diaminobenzidine (DAB) as chromogen and hematoxylin as counterstain, all from Refine HRP-Kit Leica. For double labeling IHC, sections were incubated in ER2 (30min at 100°C) followed by incubation with mouse anti-EBNA2 (clone PE2, Abcam) and detection with Refine HRP Kit Leica, without counterstaining. Prior to the second staining, slides were pretreated again for 5min at 100°C with ER2, then incubated either with mouse anti-CD30 (clone 1G12, Novocastra) or mouse anti-Pax5 (clone 1EW, Novocastra) for 60min. Detection was performed with New Fast Red as substrate and

nuclei were counterstained with hematoxylin (all from AP refine Kit, Leica). For high through-put automated analysis, the Vectra 3.0 automated quantitative pathology imaging system and inForm software from PerkinElmer were employed to analyze stained tissue sections as previously described [63]. In brief, tissue regions were identified using inForm tissue segmentation scanning protocols and images were acquired at a 20× magnification with an automated CCD camera (0.5 micron/pixel). To quantify CD30 expression, images were used to train algorithms in inForm software to recognize and count CD30, EBNA2, PAX5 single-positive and CD30/PAX5 and CD30/EBNA2 dual-positive cells, respectively. Single-stained DAB, hematoxylin or fast red control slides were used to determine the spectrum for each chromogen. The number of positive cells was determined per $1mm^2$ and 3–12 (median 7) and 6–15 (median 10) images were analyzed per section for CD30 and CD30/EBNA2, respectively.

## FK506 measurement

FK506 concentration was measured after a minimum of three applications (i.e. one week of treatment) and 24h after the last subcutaneous administration. Aliquots of whole blood were frozen and stored at −80˚C or transported at −20˚C until thawed for analysis. Sample preparation and analysis were based on a generic protein precipitation procedure followed by a tailor made liquid chromatographic separation coupled with mass spectrometry for detection. Blood samples were directly used for further preparation. Calibration, quality control, and recovery control samples were prepared by spiking blank blood with known quantities of FK506 (between 2 and 6250 ng/ml). For analyte determination, Warfarin was used as generic internal standard (IS). Aliquots of 30 μl calibration standard, quality control, recovery control, and unknown samples were transferred to 0.75 ml 96-well-plate and 3 μl IS mixture (2.5 μg/ml in 50% CH3CN) was added to each tube. For protein precipitation and extraction from the blood and brain matrix, 200 μl 80% $CH_3CN$ was added. After vortexing for 10 min, the samples were centrifuged at 3220g for 15 min at 4˚C. 50 μl of the upper layer was transferred to a 1.2 ml 96-deep-well-plate for analysis. LC–MS/MS analysis: For quantitative analysis, a 1 μl aliquot of each sample, including calibration, quality control, and recovery control samples were injected with a cooled HTS CTC PAL sample manager and Flux Rheos Alegro HPLC system. The test article and its internal standard were separated with a Phenomenex Polar RP (50 × 2.1 mm ID, 3.0 μm pore size) as column at 65˚C. For separation, a linear gradient from 30 to 99% B in 2.5 min at a flow rate of 0.400 ml/min was applied. The total cycle time was 5.0 min. The mobile phase used was A: water with 0.1% formic acid, and B: $CH_3OH$ with 0.1% formic acid. For detection the column effluent was directly guided in an AB Sciex API5500 Triple quad mass spectrometer equipped with a TurboIonSpray interface. The detection was done in MRM positive ion mode. Quantification was based on the compound/IS ratio of the extracted ion chromatograms of the selected mass transitions on the water adduct 821 m/z → 769/576 m/z for FK506 and 309 m/z → 163 m/z for Warfarin (IS). The unknown sample concentration was calculated using external calibration curves. The LLoQ of the method was set to 2 ng/ml for blood and the recovery from the matrix was 94±3%. All calculations were performed with AB Sciex Analyst software 1.6.2.

## DNA isolation and EBV DNA qPCR

Total DNA from splenic tissue or brain-derived lymphocytes and whole blood was extracted using DNeasy Blood & Tissue Kit (QIAGEN) and NucliSENS (bioMérieux), respectively, according to manufacturer's protocol. Quantitative analysis of EBV BamHI W fragment DNA was performed by TaqMan (Applied Biosystems) quantitative PCR (qPCR) as described

previously [18, 56]. Reactions were run in duplicate on an ABI Prism 7700 Sequence Detector (Applied Biosystems).

## RT qPCR of viral and host transcripts

RNA was isolated as described in "AmpliSeq transcriptome profiling" and frozen at –20˚C or immediately reverse-transcribed with GoScript Reverse Transcriptase (Promega) according to the manufacturer's recommendations using 3' gene-specific RT primers as previously described [56, 64] with the exception of EBER1 and 18S rRNA for which cDNA was generated separately using the High-Capacity cDNA Reverse Transcription Kit with random primers (Applied Biosystems). cDNA was stored at −20˚C until qPCR. Amplifications of *Cp/Wp-*, *Qp-*, *Fp—EBNA1*, *EBNA2*, *LMP1*, *LMP2A*, *BZLF1*, *GAPDH* and *SDHA* were carried out in triplicate with equal volumes of input cDNA and TaqMan universal PCR reagents (Applied Biosystems) using previously published primer sets with 5'FAM/3'TAMRA[64] or 5'FAM/3'MGB labeled probes for *SDHA* (TaqMan Applied Biosystems Gene Expression Assay (Hs00417200)). Thermal cycling was performed on a C1000 Touch CFX384 Real-Time platform (Bio-Rad) as previously described [56] and transcript levels were calculated relative to the geometric mean of the two reference genes *GAPDH* and *SDHA*. Amplification of EBER1 and 18S rRNA was performed in triplicates on the QuantStudio 12K Flex *Real-Time PCR System* using TaqMan Multiplex Master Mix (Thermo Fisher Scientific) and previously published EBER1 TaqMan assay with 5'FAM/3'QSY labeled probe [65] and 18SrRNA TaqMan assay (Applied Biosystems (#4319413E) with *5'VIC/3'MGB labeled* probe as duplex assay under standard TaqMan assay conditions. Results were normalized by the expression of 18S rRNA by using $2^{-\Delta\Delta Ct}$ calculation.

## Flow cytometric analysis

Whole blood and splenocyte cell suspensions of huNSG-A2 mice were subjected to erythrocyte lysis by ACK lysis buffer (Life Technologies) prior to staining. Fluorescently-labeled antibodies were purchased from BioLegend unless otherwise stated: CD45 (Pacific Blue/HI30), CD3 (PE/UCHT1 or BV785/OKT3), CD8 (PerCP/SK1), CD4 (APC-Cy7/RPA-T4 or BV605/OKT4), CD19 (PE-Cy7/HIB19 or AF700/HIB19), HLA-DR (FITC/L243), CD62L (APC/DREG-56, BD Bioscience) and CD45RA (BV510/HI100). Cell viability was assessed using fixable NIR and Aqua viability dyes (Zombie Fixable Viability Kit, BioLegend or LIVE/DEAD Fixable Dead Cell Stain Kit, ThermoFisherScientific). Acquisition was performed on BD FACSCanto II or BD LSR Fortessa flow cytometers and data was analyzed using FlowJo 9.9 software. Absolute leukocyte numbers were calculated from white blood cell counts measured with a hemocytometer (Beckman Coulter AcT Diff Analyzer).

## IFNγ release assay

EBV-specific T cell response was assessed by IFNγ Elispot assay for individual mice as previously described [15, 66]. In brief, splenocytes or bone marrow cells were depleted of human CD19$^+$ cells using anti-CD19 microbeads as described above. The CD19$^-$ fraction was stimulated with autologous LCLs at a ratio of 5:1 or PMA (25ng/ml) and Ionomycin (0.325μM), with FK506 (Sigma-Aldrich, 20 or 200ng/ml) or carrier solution equivalent (DMSO) for 18h in duplicate or triplicate for each huNSG mouse and spots were counted with an Elispot readersystem (ELR02, Autoimmun Diagnostika GmbH).

### Lymphocyte isolation from CNS

CNS tissues from huNSG-A2 mice were isolated as previously described [67]. In brief, animals were perfused with ice-cold PBS, CNS was removed, dissected into small pieces, suspended in digest buffer (collagenase D 0.2 mg/ml; DNAse 20 μg/ml in PBS), and incubated for 40min at 37˚C and 5% $CO_2$. The digested tissue was passed through a 70-μm nylon mesh. Cells were suspended in 30% Percoll (17-0891-01; GE Healthcare) in PBS and ultracentrifuged (Sorvall RC 6 Plus Superspeed Centrifuge; Thermo Fisher Scientific) for 30min at 4˚C. The resulting lipid layer was removed and discarded. Cellular DNA was isolated as described above.

### Human cytokine quantification and analysis

Human inflammatory cytokines in serum samples from huNSG-A2 mice or PTLD patients were measured as previously described [56]. In brief, serum was collected and frozen at −80˚C until use. Cytokine concentrations were measured in duplicates with V-PLEX Proinflammatory Panel 1 plates (Mesoscale Diagnostics, K15049D-2), read with a Meso Quickplex SQ120 (MSD) and analyzed with Discovery Workbench software V4.0.12 (Mesoscale Diagnostics). Standard dilutions of the calibrator blend for standard curve generation were performed in duplicates or quadruplicates according to the manufacturer's instructions. Heatmap generation and hierarchical clustering analysis of the cytokine data was performed using R (version 3.6.0 (2019-04-26)) and RStudio (version 1.1.463) with tidyverse (version 1.2.1, https://CRAN. R-project.org/package=tidyverse), heatmaply[68] (version: 0.16.0, empirical percentile transformation with "percetize()" and hclust_method = "ward.D2") and dedicated packages.

### Quantification of serum sCD30, PRR4 and LENEP levels

Serum was collected using BD SST Microtainer tubes (Becton Dickinson) upon terminal bleeding of animals and frozen at −80˚C until use. Serum from PTLD patients at diagnosis and after recovery and serum from healthy age-matched controls was collected and frozen at −80˚C until use. Commercially available ELISA kits (CD30 Human Instant ELISA Kit, Thermo-Fisher Scientific; lens epithelial protein (LENEP) ELISA Kit, MyBiosource; PRR4 ELISA Kit (Proline Rich 4, Lacrimal), Cloud-Clone Corp.) were used to determine the concentration of sCD30, PRR4 and LENEP in the serum of individual mice or patients and in the supernatants of cultured LCLs according to the manufacturer's instructions. Samples were measured in duplicate and plates were read on a Tecan infinite M200 PRO reader (Tecan, Maennedorf, Switzerland). The quantification thresholds were set at the mean plus three times the standard deviation of the negative controls.

## Supporting information

**S1 Data. Excel spreadsheet containing the LENEP ELISA results and the numerical data for the Figure panels 2B, 3E, 5E, S1D, S2D, S2H, S6B, S7E and S7F in separate sheet tabs.** (XLSX)

**S1 Table. Curated gene set list for GSEA.** (PDF)

**S2 Table. GSEA results.** (PDF)

**S3 Table. PTLD patients' characteristics.** (PDF)

**S4 Table. Control subjects' characteristics.**
(PDF)

**S1 Fig. Characteristics of FK506 treatment on EBV-infected animals. A)** Baseline peripheral blood human lymphocyte engraftment in huNSG-A2 mice before infection. Composite data of five independent experiments (n = 80) each reconstituted with $CD34^+$ cells from a single HFL donor (n = 5). Percentages given represent values of human $CD45^+$ lymphocytes; $CD3^+$, $CD19^+$ and $CD3^-NKp46^+$ cells within the $CD45^+$ population, and percentages of $CD4^+$ and $CD8^+$ cells of $CD3^+$ T cells. **B)** Total numbers of human $CD45^+$ cells per ml blood before infection is depicted for individual mice (n = 7–10 animals per group). **C)** Composite survival from four independent experiments is depicted for the indicated groups. Mice were euthanized when weight loss exceeded 15% of the maximum weight or when signs of morbidity necessitated a premature euthanasia based on the laboratory's animal welfare protocol (n = 15–18 animals per group). **D)** Relative weight development of mice that survived until five weeks p.i. represented as mean ± SEM percent of starting weight per group for four pooled experiments; $EBV^-FK506^-$ vs. $EBV^-FK506^+$ p = 0.6474, p = 0.4496 for week four and five respectively; $EBV^+$ $FK506^-$ vs. $EBV^+$ $FK506^+$ p = 0.0354 and p = 0.0128 for week four and five respectively (unpaired t tests). **E)** Analysis of EBV BamHI W fragment DNA detection as determined by qPCR in lymphocytes derived from the CNS after PBS perfusion at five weeks p.i.. Composite data from two independent experiments with $EBV^+$ $FK506^-$ (n = 6) and $EBV^+$ $FK506^+$ (n = 7) depicted as raw viral titers and percent of mice per group with DNA levels above the qPCR detection threshold. **F)** Blood FK506 levels (ng/ml) in EBV-infected mice with (n = 8) and without (n = 4) macroscopically visible tumors. Median, MWT p = 0.2303. **G)** EBV BamHI W DNA burden in the blood and **H)** in the spleen, measured at the day of sacrifice, in infected mice with (n = 10) and without (n = 12) macroscopically visible tumors. Median, MWT p = 0.0298 and p = 0.0004, respectively. **F-H)** Composite data from three independent experiments in which mice developed tumors. **I)** Total numbers of reconstituted human cells per ml blood prior to the infection with EBV in mice with (n = 9) and without (n = 8) tumors. Median, MWT. $CD45^+$ (p = 0.6730), $CD3^+$ (p = 0.0927), $CD3^+CD4^+$ (p = 0.1139), $CD3^+CD8^+$ (p = 0.0927), $CD19^+$ (p = 0.8148) and $NKp46^+$ (p>0.9999), from two independent experiments. **F-I)** FK506-treated mice are depicted as black symbols, untreated mice as clear symbols. *: p<0.05, ***: p<0.001.
(TIF)

**S2 Fig. Peripheral blood and spleen T cell frequencies and phenotype. A)** The numbers of total $CD4^+$ or $CD8^+$ T cells in the blood at baseline and 5 weeks p.i. are depicted for individual mice of two independent experiments (n = 7–10 animals per group), Wilcoxon signed-rank test. **B)** Total numbers of T cell subsets in the spleen (left) and blood (right) in mice from a representative experiment with n = 3–6 animals per groups. Median. **C)** The number of total $CD4^+$ and $CD8^+$ T cells in the spleen is plotted relative to the mean total $CD4^+$ or $CD8^+$ T cells of the $EBV^-FK506^-$ group of each HPC donor. Median (IQR), MWT. **D)** Differentiation status of T cells in the spleen is represented as stacked bar graphs (n = 7–10 animals per group). Naïve: $CD62L^+$ $CD45RA^+$, central memory: $CD62L^+$ $CD45RA^-$, effector memory: $CD62L^-$ $CD45RA^-$, effector memory $RA^+$ (EMRA): $CD62L^-$ $CD45RA^+$. **E)** Relative counts of activated $CD4^+$ or $CD8^+$ T cells in the spleen were determined by HLA-DR$^+$ surface staining. Median (IQR), MWT. **F-G)** Relative counts of total and activated (HLA-DR$^+$) $CD4^+$ and $CD8^+$ T cells are depicted for **F)** the blood and **G)** the spleen of tumor-bearing and tumor-free mice of the $EBV^+$ $FK506^+$ group from three independent experiments. Cell numbers are presented relative to the respective mean cell count of the $EBV^-FK506^-$ group of each HPC donor. Tumor presence in individual mice is indicated by symbol color: clear = no tumors, black = 1 tumor,

red = 2 or more tumors. Median (IQR), MWT. **H)** IL-2 concentration (pg/ml) was measured in the serum obtained on the day of sacrifice. EBV⁻FK506⁺ (n = 5), EBV⁻FK506⁺ (n = 8), EBV⁺ FK506⁻ (n = 6), EBV⁺ FK506⁺ (n = 10). Median (IQR), MWT. **I)** Blood CD4⁺ and CD8⁺ T cell counts in PTLD patients and controls (Ctrl) derived from data previously reported by Sebelin-Wulf et al. [26]. Healthy EBV positive controls (EBV⁺ HD, n = 6), EBV positive transplant recipients without PTLD development (EBV⁺ no PTLD, n = 2), PTLD patients with histologically verified EBV association (EBV⁺ PTLD, n = 4). Median (IQR), MWT. **J)** Correlations between the relative number of splenic CD8⁺ T cells and the splenic EBV DNA load are depicted for FK506-treated and non-treated EBV-infected mice. r, spearman correlation. Solid lines represent trend lines obtained by linear regression and shaded areas indicate 95% CI of each trend line. **K)** IFNγ response was assessed by Elispot assay whereby IFNγ spot-forming cells (SFC) were counted with and without stimulation of CD19 MACS-depleted splenocytes with PMA/Ionomycin and *in vitro* supplementation of FK506 at 200ng/ml to cells for individual EBV-infected mice with FK506-treatment (n = 6). Composite data from two (**A, D, H**), three (**F**) or four (**C, E, G, J**) independent experiments. *: p<0.05, **: p<0.01, ***: p<0.001.
(TIF)

**S3 Fig. EBV mRNA expression in CD19⁺ B cells.** The number of CD19⁺ cells in **A)** blood and **B)** spleen is depicted for tumor-bearing and tumor-free mice of the EBV⁺ FK506⁺ group from three independent experiments. CD19⁺ cell counts are presented relative to the mean CD19⁺ cell count of the EBV⁻FK506⁻ group of each HPC donor. Median (IQR), MWT. **C-D)** EBV transcript expression as measured by RT-qPCR in CD19⁺ splenocytes is depicted for individual mice from three independent experiments with EBV⁺ FK506⁻ (n = 9) and EBV⁺ FK506⁺ (n = 11). **C)** *EBER1* expression relative to the *18S rRNA* reference gene and normalized per donor. **D)** *Cp/Wp-EBNA1*, *EBNA2* and *LMP1* and *-2A* transcript expression normalized to *EBER1*. Median, MWT. **A-D)** Tumor presence in individual mice is indicated by symbol color: clear = no tumors, black = 1 tumor, red = 2 or more tumors.
(TIF)

**S4 Fig. Characteristics of mice included in gene expression profiling of B cells. A)** EBV DNA load in whole blood and **B)** in the spleen of mice at the day of sacrifice. **C)** Percent of all mice included in the AmpliSeq RNA profiling with macroscopically visible tumors with absolute numbers of mice indicated per group. **D)** Total numbers T cells and **E)** activated T cells per spleen normalized per donor. **F)** Total splenic CD19⁺ cell counts normalized per donor. **A-B, D-F)** Black dots indicate individual mice included in the RNA profiling analysis. Grey boxplots indicate median (IQR) and range of the total respective data set presented in Figs 1, 3 and S2. **G)** EBV transcript expression relative to the geometric mean of *GAPDH* and *SDHA* levels. *EBER1* expression relative to the *18S rRNA* reference gene. Mice included in the RNA profiling analysis are depicted as black symbols; other mice of the total respective data set presented in Figs 3 and S3 are depicted as grey symbols.
(TIF)

**S5 Fig. Heatmap of the top 100 DEGs.** Heatmap of the gene expression vs. sample matrix from RNA expression profiling of the human B cell transcriptome of mice from three independent experiments with EBV⁻FK506⁻ (n = 5, clear), EBV⁻FK506⁺ (n = 6, orange), EBV⁺ FK506⁻ (n = 7, purple) and EBV⁺ FK506⁺ (n = 8, magenta). Displayed are the top 100 differentially expressed genes (DEGs) of all samples (i.e. individual mice). Hierarchical clustering separates genes with positive log fold change from negative and tries to group different sample

types.
(TIF)

**S6 Fig. Characteristics of gene expression in B cells from EBV-infected huNSG-A2 mice treated with FK506. A)** Venn diagrams depicting the number of enriched gene sets for the four comparisons EBV$^+$ FK506$^-$ vs. EBV$^-$FK506$^-$ and EBV$^+$ FK506$^+$ vs. EBV$^-$FK506$^+$ (upper panel), and EBV$^-$FK506$^+$ vs. EBV$^-$FK506$^-$ and EBV$^+$ FK506$^+$ vs. EBV$^+$ FK506$^-$ (lower panel). Selected up-regulated gene sets are indicated to the right, down-regulated gene sets to the left. **B)** IFNγ concentration (pg/ml) was measured in the serum obtained on the day of sacrifice. EBV$^-$FK506$^-$ (n = 5), EBV$^-$FK506$^+$ (n = 8), EBV$^+$ FK506$^-$ (n = 6), EBV$^+$ FK506$^+$ (n = 10). Mean ± SEM, MWT. ***: p<0.001.
(TIF)

**S7 Fig. Serum PRR4 concentration in FK506-treated huNSG-A2 mice and PTLD patients, sCD30 in pediatric controls and soluble serum factor production by LCLs *in vitro*. A-C)** Protein concentration of PRR4 was measured in serum samples with ELISA. **A)** Serum was obtained at the day of sacrifice from animals of the indicated groups and are depicted from two independent experiments. Dashed lines indicate quantification thresholds. Mean ± SEM, MWT. **B)** *Post hoc* stratification of FK506-treated mice with or without macroscopically visible tumors with EBV-infection status indicated by black (EBV$^+$) or clear (EBV$^-$) symbols. Mean ± SEM, MWT. **C)** PRR4 was measured in the sera of 8 pediatric PTLD patients at diagnosis and again upon recovery at least 36 months later (see S3 Table). Wilcoxon matched-pairs test. **D)** sCD30 was measured in the sera of 31 healthy children undergoing elective tonsillectomy and 21 children that presented with non-EBV associated fever (Febrile non-EBV, EBV VCA-IgM negative). Approximately 50% of each of these control cohorts were positive for EBV VCA-IgG (see S4 Table). Median (IQR), MWT. **E)** Concentrations of sCD30 and **F)** IL-6 and IL-10 were measured in the supernatant of LCLs obtained by the *ex vivo* expansion of cells from individual EBV-infected humanized mice (n = 4). LCLs were seeded at 5000 or 10000 cells and supernatant was collected after 0, 3 and 7 days of culture. **: p<0.01.
(TIF)

## Acknowledgments

The authors would like to thank past and present members of the Viral Immunobiology Lab at the Institute of Experimental Immunology, UZH, for technical support. Further, the authors would like to thank Peter Wipfli and Rowan Stringer for bioanalytical support, and Brigitte Christen for RNA quality and quantity measurements.

## Author Contributions

**Conceptualization:** Nicole Caduff, Donal McHugh, Elisabetta Traggiai, Michael Kammüller, Christian Münz.

**Data curation:** Nicole Caduff, Donal McHugh, Michael Prummer.

**Formal analysis:** Nicole Caduff, Donal McHugh, Michael Prummer.

**Funding acquisition:** Nicole Caduff, Donal McHugh, Michael Kammüller, Christian Münz.

**Investigation:** Nicole Caduff, Donal McHugh, Anita Murer, Patrick Rämer, Ana Raykova, Vanessa Landtwing, Lisa Rieble, Christian W. Keller, Laurent Hoffmann, Friedrich Raulf.

**Methodology:** Nicole Caduff, Donal McHugh, Michael Prummer, Christian Münz.

**Project administration:** Nicole Caduff, Donal McHugh, Christian Münz.

**Resources:** Janice K. P. Lam, Alan K. S. Chiang, Tarik Azzi, Christoph Berger, Tina Rubic-Schneider, Elisabetta Traggiai, Michael Kammüller, Christian Münz.

**Software:** Nicole Caduff, Donal McHugh, Michael Prummer.

**Supervision:** Tina Rubic-Schneider, Elisabetta Traggiai, Jan D. Lünemann, Michael Kammüller, Christian Münz.

**Validation:** Nicole Caduff, Donal McHugh, Michael Prummer.

**Visualization:** Nicole Caduff, Donal McHugh, Michael Prummer.

**Writing – original draft:** Nicole Caduff, Donal McHugh, Christian Münz.

**Writing – review & editing:** Nicole Caduff, Donal McHugh, Anita Murer, Patrick Rämer, Ana Raykova, Vanessa Landtwing, Lisa Rieble, Christian W. Keller, Michael Prummer, Laurent Hoffmann, Janice K. P. Lam, Alan K. S. Chiang, Friedrich Raulf, Tarik Azzi, Christoph Berger, Tina Rubic-Schneider, Elisabetta Traggiai, Jan D. Lünemann, Michael Kammüller, Christian Münz.

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
