## [Decision Letter · Decision Letter 0]

24 Sep 2019

Dear Dr. Muenz,

Thank you very much for submitting your manuscript "Immunosuppressive FK506 treatment exacerbates EBV-associated lymphoproliferative disease in humanized mice" (PPATHOGENS-D-19-01491) for review by PLOS Pathogens. Your manuscript was fully evaluated at the editorial level and by three experts in the EBV field. The reviewers appreciated the attention to an important problem, but raised some substantial concerns about the manuscript as it currently stands. These issues (particularly from Reviewer #1 and #3) must be addressed before we would be willing to consider a revised version of your study. We cannot, of course, promise publication at that time.

We therefore ask you to modify the manuscript according to the review recommendations before we can consider your manuscript for acceptance. Your revisions should address the specific points made by each reviewer.

(1) A letter containing a detailed list of your responses to the review comments and a description of the changes you have made in the manuscript. Please note while forming your response, if your article is accepted, you may have the opportunity to make the peer review history publicly available. The record will include editor decision letters (with reviews) and your responses to reviewer comments. If eligible, we will contact you to opt in or out.

(2) Two versions of the manuscript: one with either highlights or tracked changes denoting where the text has been changed; the other a clean version (uploaded as the manuscript file).

Additionally, to enhance the reproducibility of your results, PLOS recommends that you deposit your laboratory protocols in protocols.io, where a protocol can be assigned its own identifier (DOI) such that it can be cited independently in the future. For instructions see http://journals.plos.org/plospathogens/s/submission-guidelines#loc-materials-and-methods

We hope to receive your revised manuscript within 60 days. If you anticipate any delay in its return, we ask that you let us know the expected resubmission date by replying to this email. Revised manuscripts received beyond 60 days may require evaluation and peer review similar to that applied to newly submitted manuscripts.

[LINK]

Sincerely,

Zhen Lin

Associate Editor

PLOS Pathogens

Erik Flemington

Section Editor

PLOS Pathogens

Kasturi Haldar

Editor-in-Chief

PLOS Pathogens

orcid.org/0000-0001-5065-158X

Grant McFadden

Editor-in-Chief

PLOS Pathogens

orcid.org/0000-0002-2556-3526

Reviewer's Responses to Questions

**Part I - Summary**

Reviewer #1: This manuscript uses a humanized mouse model to examine whether FK506 treatment (a calcineurin inhibitor often used to immunosuppress transplant patients) increases the ability of EBV infection to induce lymphomas in vivo. It has been previously shown by other groups that various types of agents that inhibit T cell function increase the ability of EBV to induce lymphomas in humanized mice, since T cells have some ability to control EBV in these models. Not surprisingly, the authors find that EBV-infected mice treated with FK506 (a known T cell function inhibitor) develop more frequent tumors and have higher viral loads. RNAseq analysis in EBV-infected animals treated with and without FK506 identifies CD30 (a cellular gene already well known to be stimulated by EBV infection of B cells, and already shown to be increased in the plasma of patients with mononucleosis and PTLD) as being increased in the EBV-infected mice treated with FK506. The authors then confirm that levels of soluble CD30 may be a good biomarker for detecting EBV+ PTLD in transplant patients.

Over-all the study is well performed and the data are of high quality. However, the results are not particularly novel or surprising and provide only incremental advances to previously published literature.

Reviewer #2: In this interesting manuscript the authors develop a model of Epstein Barr virus associated post-transplant lymphoproliferative disease (EBV-LPD) in humanized mice. They show, using state of the art technologies, the effect of tacrolimus (FK506)-mediated immunosuppression. While the current conclusions are not particular novel, the model represents a significant advance since it will allow for the first time the proper preclinical modeling of this complication post hematopoietic and solid organ transplant. Thus, it needs to be considered a significant advance in the field.

The authors studies are comprehensive and are well designed. Data sets are analyzed meticulously and are experiments have appropriate controls. The conclusions of the authors are solid and are supported by the presented data. It is one of the few manuscripts I have reviewed within the last 5 years for which I have no concerns.

Reviewer #3: This article presents a large variety of interesting and detailed observations and data regarding the biological consequences of experimental EBV infection in a humanized mouse model after application of the calcineurin inhibitor FK506.

Since only a minority of animals develops tumors, it may be that this model is not predominantly a model of PTLD, depending on how that term is defined. Nontheless, the model has many interesting features.

Since only some animals had tumors, it would be interesting to learn more about the correlates and parameters associated with tumor formation, based on the data already available to the authors. The authors have analyzed a multitude of parameters regarding immune cell reconstitution and activation, viral titers, viral and cellular gene expression etc. Not only tumor formation but also some other of these parameters vary quite widely within a treatment group, especially the EBV+FK506 group. It would be nice to learn more about how these parameters co-varied within a group. Maybe the authors can find ways to present their data that make such correlations more transparent.

**Part II – Major Issues: Key Experiments Required for Acceptance**

Reviewer #1: (No Response)

Reviewer #2: As outlined in the summary, believe that the developed model is a great step forward in the preclinical modeling of EBV-LPD. I have no major or minor concerns with the submitted manuscript.

Reviewer #3: (No Response)

**Part III – Minor Issues: Editorial and Data Presentation Modifications**

Reviewer #1: (No Response)

Reviewer #2: None

Reviewer #3: Individual remarks

title, "exacerbates" - this title does not seem to fit the observations very well. The authors found that FK506 leads to "elevated viral burden, more frequent tumor formation and diminished EBV-induced T cell responses" - but this is not the same as saying that PTLD was exacerbated, as opposed to raised in incidence. For example, it is not documented whether tumors were larger in FK506-treated animals. Weight loss was more pronounced; virus titers were higher; but B cell number was not increased.

p71, "with" missing

Fig. 1B, does it show FK506 levels 24h after the *first* administration of FK506?

Fig. S1A shows only percentages. How good was human cell reconstitution really, i.e., what were absolute numbers of hCD45+ cells?

Fig.S1D, not many mice were EBV-positive, it would be better to represent the viral titers and not just positive vs. negative.

Where were the tumors located? What was their size? Is other information available about them?

Since only a minority of mice got tumors (Fig. 1E), it would be interesting and important to know whether these were mice that had

- higher B cell counts than others, or lower T or NK cell counts at infection?

- higher FK506 plasma levels?

- higher EBV load in blood or spleen?

- same or different hHSC source samples than non-tumor mice?

The answer to these questions could, for example, in part be provided by representing the mice with 0, 1, or >1 tumors by different symbols in Fig. 1B,C,D and Fig S1A,D.

Fig. 2A,B,C,D,F, S2B,C,D,G,H, please show the actual cell numbers instead of normalized "fold over mock". That's more informative.

Fig.2E shows evidence of T cell "attenuation" due to FK506, but only in the Mock situation (not the EBV situation). What is the p value for this effect? It's missing in the diagram.

line 138, "a similar effect ... (Fig. S2B)", maybe, but not significant in this case, please amend.

line 140, "whereby more central- or effector memory T cells were found", yes, but only in some of the comparisons, e.g. in Fig. 2B, CD4+, mock-FK506 vs EBV-FK506, there was no difference. The following sentence ("Yet, in...") does not really clarify this.

line 143, "clearly diminished", again, this was seen only for some of the conditions shown, not for CD8+EBV±FK506 in blood and spleen (Fig 2C and S2D) and not for CD4+ in the same condition in spleen (S2D). The following sentence ("However,...") seems to acknowledge this in some way, but is not precise enough (CD4+EBV spleen were not reduced by FK506), and also seems to partially contradict the previous sentence. Overall, the description in the text is not clear enough here.

line 147, "steady state conditions" seems to be a synonym for mock infection but is not a very suitable one, since the mock-infected mice were not that steady in their state either (see S1B,C, mortality and weight loss).

line 150-164, the situation is quite complex, and so is the present interpretation. The argument in favor of curbed CD8+ responses is sophisticated but not entirely convincing. Major findings were that CD8+ responses and EBV-specific responses were consistent in being not significantly reduced by FK506 in the EBV situation (2A,C,E, S2B,D). In spite of this, the authors seem to suggest that the "positive correlation between activated or total CD8+ T cells and the rate of IFNγ-secreting T cells in EBV-infected animals but not in those treated with FK506 (Figs 2F and S2H)" is an argument in favour of specific T cell attenuation. However, since in the FK506 situation (2F, right; S2H, right) the sample number was smaller (n=6), it was more likely to result in non-significance. It is always difficult to draw positive conclusions from the absence of significance, especially if the sample size is small.

Fig. 3A, why were two different statistical tests used in this single analysis?

Fig. 3CD (also Fig. 1D), these data span several orders of magnitude and are accordingly represented on a log scale. However, their arithmetic mean is presented. The geometric mean or median would fit the structure of the data better. The arithmetic mean is unduly influenced by isolated high datapoints, as can be seen e.g. in Fig.3D LMP1 EBV. Since the non-parametric Mann-Whitney test was used, the median would probably be the most adequate.

line 176, "equalized", this seems to simplify the real situation. In fact, normalization to EBER1 made it even clearer that three values stood out in the EBV+FK506 condition for each of the three transcripts EBNA2, LMP1, LMP2, and maybe even EBNA1 Cp/Wp. It seems likely that these three samples are the same for the different transcripts? If correct, it would be good to let readers know. It would be interesting to know whether the latency III pattern was synchronously induced in some samples but not others, in what looks like a bimodal distribution. Were some of the "latency III high" mice carrying tumors?

Fig. 4, since the number of samples that underwent this gene expression analysis was necessarily limited (4,5,6,7 samples for the four conditions), it seems important how the samples were chosen. Were these some of the same mice for which data were shown in Fig. 1-3? Which ones? Did the EBV samples include mice with tumors, and how many? Did the samples include mice that had high and low levels of latency III genes (see above, Fig. ), or high and low levels of EBV replication, or high and low levels of CD8 T cells, or activated T cells etc.? If this information is available, it should be revealed to get a better idea how representative these data are of the treatment groups.

line 203, "This is likely due...", I don't quite understand this explanation.

line 362 "four or five weeks post infection", but Fig. 1 displays "Day 35" as scheduled day of sacrifice if mice needn't be euthanized earlier. This seems inconsistent, please clarify.

line 367 "were excluded from the EBV group", but hopefully not included in the non-EBV groups, therefore excluded altogether?

line 371 on B cell purification and RNA isolation, what was the minimum number of CD19+ B cells per sample that was analyzed?

line 386, "library size ... was normalized to 1 million" - i.e. reads? How many actual reads were acquired per sample at a minimum?

PLOS authors have the option to publish the peer review history of their article (what does this mean?). If published, this will include your full peer review and any attached files.

Reviewer #1: No

Reviewer #2: No

Reviewer #3: No

---

## [Decision Letter · Decision Letter 1]

13 Feb 2020

Dear Dr. Muenz,

Thank you very much for submitting your manuscript "Immunosuppressive FK506 treatment leads to more frequent EBV-associated lymphoproliferative disease in humanized mice" for consideration at PLOS Pathogens.

Your revised manuscript has been evaluated by 4 experts. Most of the reviewers believe that this is an important study and should be published in the PLoS Pathogens. Some additional concerns were raised by Reviewer 4, please address them. 

Based on the reviews, we are likely to accept this manuscript for publication, providing that you modify the manuscript according to the review recommendations.

Sincerely,

Zhen Lin

Associate Editor

PLOS Pathogens

Erik Flemington

Section Editor

PLOS Pathogens

Kasturi Haldar

Editor-in-Chief

PLOS Pathogens

orcid.org/0000-0001-5065-158X

Michael Malim

Editor-in-Chief

PLOS Pathogens

orcid.org/0000-0002-7699-2064

Your revised manuscript has been evaluated by 4 experts. Most of the reviewers believe that this is an important study and should be published in the PLoS Pathogens. Some additional concerns were raised by Reviewer 4, please address them.

Reviewer Comments (if any, and for reference):

Reviewer's Responses to Questions

**Part I - Summary**

Reviewer #1: This paper examines the effect of an immunosuprressive NFAT inhibitor (FK506), used to inhibit T cell function in transplant patients, on the ability of EBV to induce lymphomas in a humanized mouse model. Not surprisingly, inhibiting T cell function results in increased lymphomas. This has previously been shown to be the case in EBV-infected humanized mouse models when T cells are depleted. The authors also show that a known EBV target cellular gene (CD30) is increased in animals with EBV induced tumors. As stated in my previous review, the results are not particularly novel. The revised manuscript does not change this observation.

Reviewer #2: The authors have developed a preclinical model to study EBV-PTLD in HLA-A2 transgenic humanized mice. They have performed very thorough studies on analyzing immune responses to EBV in these mice. While the current findings cannot be considered novel (soluble CD39), believe that the model will be of great value in the future as more targeted therapies to treat PTLD are being developed. In addition, believe that it will be of high interest to scientist, who study EBV and EBV-specific immune responses.

Reviewer #3: The authors have very thoroughly and successfully revised their manuscript. I am indebted to the authors for their detailed explanations and additional information. I have no further requests.

Reviewer #4: This revised manuscript by McHugh et al described the use of the CD34 humanized mouse model to study the effects of EBV infection and FK506 treatment as a model for post-transplant lymphoproliferative disease (PTLD) in humans. The authors have demonstrated that the humanized mice infected with EBV and treated with FK506 have compromised T cell immune responses, leading to higher EBV viral load, and higher risks in developing tumors. The authors further analyzed the transcriptomes of the treated and infected groups with controls to identify the differentially expressed genes, and identify soluble CD30 as a potential marker for tumor development in the humanized mice and PTLD. This revised manuscript have addressed most of the concerns raised by the previous reviews and have added new experiments and analysis of the data. The authors have also re-written parts of the manuscripts to clarify some of the points that were made. The manuscript, as indicated by the previous review, that the study was well designed and conducted, the resulting immunosuppression resulting tumor development in the animals are expected, and represented incremental knowledge gain in the field. The novelty nevertheless, is the use of the humanized mouse model for EBV infected and FK506 treatment, and the identification of sCD30 as potential biomarker for PTLD. There are however, a few remaining points that the authors should address

**Part II – Major Issues: Key Experiments Required for Acceptance**

Reviewer #1: (No Response)

Reviewer #2: none

Reviewer #3: (No Response)

Reviewer #4: The demonstration of soluble CD30 as potential biomarker in PTLD patients to confirm the animal model results is one of the most significant findings of the study. However, while the results were statistically significant when comparing the sCD30 levels between diagnosis and recovery, the sample size (n=8) is still small. Several of the 8 samples analyzed have low levels initially and the decrease seems insignificance. Thus, a larger sample size needs to be analyzed. Moreover, more importantly, the study lacks proper control, the levels in normal non-disease individual needs to be determined.

**Part III – Minor Issues: Editorial and Data Presentation Modifications**

Reviewer #1: (No Response)

Reviewer #2: none

Reviewer #3: (No Response)

Reviewer #4: The authors did not fully addressed the former review’s concern on the need to analyze the data for the EBV infected and treated mice that developed tumors versus those that did not, as an additional variable. The authors have provided information in comparing animals with and without tumors in figure 1 but have not compare their differences in immune cell populations and transcriptome profiles. There were a number of animals that were infected and treated with FK506 did not develop tumors but have similar viral loads in spleen and blood as those that developed tumors, the differences in a majority of the infected and treated animals that developed tumors versus those that did not need to be investigated and analyzed further.

PLOS authors have the option to publish the peer review history of their article (what does this mean?). If published, this will include your full peer review and any attached files.

Reviewer #1: No

Reviewer #2: No

Reviewer #3: No

Reviewer #4: No
---

## [Editor Report · Decision Letter 2]

15 Mar 2020

Dear Dr. Muenz,

We are pleased to inform you that your manuscript 'Immunosuppressive FK506 treatment leads to more frequent EBV-associated lymphoproliferative disease in humanized mice' has been provisionally accepted for publication in PLOS Pathogens.

Best regards,

Zhen Lin

Associate Editor

PLOS Pathogens

Erik Flemington

Section Editor

PLOS Pathogens

Kasturi Haldar

Editor-in-Chief

PLOS Pathogens

orcid.org/0000-0001-5065-158X

Michael Malim

Editor-in-Chief

PLOS Pathogens

orcid.org/0000-0002-7699-2064
---

## [Editor Report · Acceptance letter]

30 Mar 2020

Dear Dr. Muenz,

We are delighted to inform you that your manuscript, "Immunosuppressive FK506 treatment leads to more frequent EBV-associated lymphoproliferative disease in humanized mice," has been formally accepted for publication in PLOS Pathogens.

Best regards,

Kasturi Haldar

Editor-in-Chief

PLOS Pathogens

orcid.org/0000-0001-5065-158X

Michael Malim

Editor-in-Chief

PLOS Pathogens

orcid.org/0000-0002-7699-2064